# Distributional Reward Estimation for Effective Multi-Agent Deep Reinforcement Learning

**Jifeng Hu[1]**    **Yanchao Sun[2]**    **Hechang Chen[3]***    **Sili Huang[4]**

**Haiyin Piao[5]**    **Yi Chang[6]**    **Lichao Sun[7]***

[1,3,4,6]School of Artificial Intelligence, Jlilin University, Changchun, China
[2]Department of Computer Science, University of Maryland, College Park, MD 20742, USA
[5]Northwestern Polytechnical University, Xian, China
[7]Lehigh University, Bethlehem, Pennsylvania, USA
[1,4]`hujf21@mails.jlu.edu.cn, huangsl21@mails.jlu.edu.cn`
[2]`ycs@umd.edu`
[3,6]`chenhc@jlu.edu.cn, yichang@jlu.edu.cn`
[5]`haiyinpiao@mail.nwpu.edu.cn`
[7]`lis221@lehigh.edu`

## Abstract

Multi-agent reinforcement learning has drawn increasing attention in practice, e.g., robotics and automatic driving, as it can explore optimal policies using samples generated by interacting with the environment. However, high reward uncertainty still remains a problem when we want to train a satisfactory model, because obtaining high-quality reward feedback is usually expensive and even infeasible. To handle this issue, previous methods mainly focus on passive reward correction. At the same time, recent active reward estimation methods have proven to be a recipe for reducing the effect of reward uncertainty. In this paper, we propose a novel Distributional Reward Estimation framework for effective Multi-Agent Reinforcement Learning (DRE-MARL). Our main idea is to design the multi-action-branch reward estimation and policy-weighted reward aggregation for stabilized training. Specifically, we design the multi-action-branch reward estimation to model reward distributions on all action branches. Then we utilize reward aggregation to obtain stable updating signals during training. Our intuition is that consideration of all possible consequences of actions could be useful for learning policies. The superiority of the DRE-MARL is demonstrated using benchmark multi-agent scenarios, compared with the SOTA baselines in terms of both effectiveness and robustness.

## 1 Introduction

Multi-agent reinforcement learning (MARL) has achieved substantial successes in solving real-time competitive games [30], robotic manipulation [1], autonomous traffic control [32], and quantitative trading strategies [15]. Most existing works require agents to receive high-quality supervision signals, i.e., rewards, which are either infeasible or expensive to obtain in practice [61]. The rewards provided by the environment are subject to multiple kinds of randomness. For example, the reward collected from sensors on a robot will be affected by physical conditions such as temperature and lighting, which makes the reward full of bias and intrinsic randomness. The interplay among agents will result in more reward uncertainty. For instance, though an agent executes the same action under the same state, the reward can still vary because other agents can execute other actions. Thus, handling non-

---

*Corresponding Author. Email address: chenhc@jlu.edu.cn (H. Chen) and lis221@lehigh.edu (L. Sun).

stationary rewards in the learning process is a necessity for successfully learning complex behaviors in multi-agent environments.

In fact, there exist numerous works [23, 28, 61, 60] in reinforcement learning (RL) that consider the *reward uncertainty* during training. The traditional stream of research on non-stationary feedback focuses on passive reward correction. For example, Wang et al. [60] adopt the confusion matrix to model the reward uncertainty and obtain a stable reward for learning. While in [61], the authors recover the true supervision signals with peer loss, which punishes over-agreement for avoiding overfitting. However, the assumption of the reward uncertainty of the above works is restrictive. For instance, the reward flipping mechanism [60] restricts the randomness of the reward to a countable value set. Recently, another stream of research focuses on active reward estimation. To alleviate the reward uncertainty issue, some works have treated reward estimation as a point-to-point regression problem where each state-action pair is mapped to a reward [38, 64]. However, the reward uncertainty, particularly caused by agents' interplay, can not be fully resolved by the point-to-point reward estimation in MARL. This is because the regression is good at the one-to-one mapping between state and rewards. But in MARL, the same state-action pair input of one agent will lead to multiple environmental rewards (i.e., one-to-many mapping), which is intractable for regression and thus hurts the performance. Moreover, these methods do not consider the fact that the reward uncertainty comes not only from inherent environment randomness but also from the interplay among agents. These two factors cause blended influence on the received reward and increase the training difficulty. As we experimentally show later (See Figure 3 (left) for details.), such a point-to-point strategy leads to a worse suboptimal outcome with the increasing agent number and the reward uncertainty degree [31].

In this work, we aim to develop distributional reward estimation followed by policy-weighted reward aggregation for MARL. Intuitively, our idea is like one human constructs a reward blueprint of all action branches in his brain and thoughtfully makes a decision by considering all possible consequences. Traditional methods [27, 41] evaluate the policy only with environment rewards, which will bring about more uncertainty when training the critic because they only take into account the reward $r_k$ received after taking a specific action, i.e., $k$-th action $a_k$, from the policy. While in our method, we consider not only the environmental rewards but also the potential rewards on other action branches to perform more stable critic updating and thus achieve better performance. So we propose multi-action-branch distributional reward estimation to model reward distributions $\{\tilde{R}^i(o^i, a_k^i)\}_{k=1}^K$ on all action branches, where $o_t^i$ is the observation and $a_{k,t}^i$ is the $k$-th action. Then we aggregate the environment reward $r_k$ and rewards sampled from different action-branch distributions $\{\tilde{R}^i(o^i, a_m^i)\}_{m=1, m \neq k}^K$ by weighting them according to the corresponding action selection probability of the current policy. We will obtain the aggregated mix reward $\bar{R}^i$ and lumped reward $\bar{r}^i$ for training each agent through the reward aggregation. Reward aggregation enables the agent to evaluate its historical decisions thoughtfully, thus providing a sophisticated and effective way to reduce the influence of reward uncertainty. Such a model covers the reward uncertainty in MARL environments and achieves better performance in several MARL benchmarks from small to large agent numbers.

Our contribution is three-fold. 1) We propose a novel framework, called Multi-agent Distributional Reward Estimation (DRE-MARL), to systematically characterize the reward uncertainty in MARL by modeling reward distributions for all action branches. To the best of our knowledge, this is the first effort to solve the reward uncertainty of MARL with multi-action-branch distributional reward estimation followed by reward aggregation. 2) Policy-weighted reward aggregation is developed in our framework which enables us to perform stable training of the critic and actors. Besides, DRE-MARL is a universal framework that can be expediently integrated into other MARL methods. 3) We validate the performance of our algorithm with function approximation and mini-batch update via extensive simulations in MARL scenarios with different agent numbers and reward uncertainty.

## 2   Related Work

**Reward Uncertainty.**   Dealing appropriately with reward uncertainty has received quite a bit of attention in recent reinforcement learning studies [61, 60, 29, 16, 47, 24, 17, 28, 42, 59]. The core idea of this line of works is to assume access to the knowledge of the noise and define unbiased surrogate loss functions to recover the true loss or reward. A typical seminal work dates back to [28], which recovers the true loss from the noisy label distribution with the knowledge of the noise rates of labels. Follow-up works offer solutions to estimate the uncertainty level from model predictions [3, 65, 67,

52, 47] or clusterable representations [66]. A couple of recent works [60, 67, 29] have also looked into this problem in sequential settings. For example, Everitt et al. [6] analyze the potential sources of uncertainty and provide the impossibility result for training under facultative reward uncertainty. Wang et al. [60] consider modeling the reward uncertainty being caused by a confusion matrix and design a statistics-based estimation method to cover the uncertainty. Wang et al. [61] recover the true supervision signals with peer loss, which punishes over-agreement to avoid overfitting. But the method is inefficient in MARL, because the uncertainty is bigger than single agent RL.

**Reward Estimation.**   A number of prior literature in robotic and non-robotic domains have adopted reward estimation [8, 38, 7, 44, 11, 55, 35, 56, 9, 51, 46, 57, 50, 25]. In most of these cases, the predicted reward is used for planning. They take advantage of the estimated reward in imagination augmented rollouts with a dynamic model (i.e., world model) accompanying the reward estimator. However, in our case, we aim at handling reward uncertainty rather than planning with multi-action-branch reward estimation and aggregation, thus avoiding learning system dynamics and multi-step imaginary rollouts. Reward estimation can also be performed by reward shaping [63, 12] (RS) and inverse reinforcement learning [10, 18] (IRL). But IRL doesn't account for the reward uncertainty, and RS focuses on efficient exploration. Recently, Romoff et al. [38] proposes to train reward estimator with function approximation alongside the value function. However, it adopts point-to-point reward estimation, which struggles when generalizing to multi-agent settings.

**Distributional RL.**   Distributional RL has recently gained increasing attention due to its powerful capacity for treating RL's uncertainty [48, 49, 34, 37, 2, 39, 33, 5]. Early studies about return uncertainty can be dated back to Sobel [45]. In MARL, the uncertainty issue is a more challenging problem than single RL. Fortunately, many studies focus on it [48, 49, 34]. For example, Sun et al. [48] propose the Mean-Shape Decomposition method and quantile mixture in value decomposition, bridging the gap between distributional RL and value function factorization. Although our work also belongs to the distributional field, we focus on the reward uncertainty of the environment rather than Q-value uncertainty and model the distribution of reward rather than Q compared with [48, 49, 34]. Besides, we propose reward estimation followed by reward aggregation, which is novel and effective.

## 3    Preliminaries

A Markov game [20, 53], a multi-agent extension of the Markov decision process, with partial observability can be described as the tuple $\mathcal{M} = \langle \mathcal{N}, \mathcal{S}, \{\mathcal{A}^i\}_{i \in \mathcal{N}}, \{\mathcal{O}^i\}_{i \in \mathcal{N}}, \mathcal{P}, \{R^i\}_{i \in \mathcal{N}}, \gamma \rangle$, where $\mathcal{N} = \{1, ..., N\}$ is the set of $N$ agents, $\mathcal{S}$ is the state space of the environment, $\mathcal{A}^i$ is the action space of agent $i$, $\mathcal{O}^i$ is the observation space of agent $i$, $\mathcal{P} : \mathcal{S} \times \mathcal{A}^1 \times \cdots \times \mathcal{A}^N \to \Delta(\mathcal{S})$ ($\Delta(x)$ represents the space of distributions over $x$.) denotes the state transition probability that is a mapping from the current state and the joint action to the probability distribution over the state space, and $\gamma \in [0, 1)$ is the discounting factor. We consider the fact that the interplay between agents results in distributional reward feedback: instead of observing $R^i$ for each agent $i$, agent $i$ can only observe the reward sampled from the reward distribution $\widehat{R}^i : \mathcal{S} \times \mathcal{A}^1 \times \cdots \times \mathcal{A}^N \to \Delta(\mathbb{R})$.

At each time t, $N$ agents receive different observations $(o_t^1, ..., o_t^N)$ and output certain joint action $\boldsymbol{a_t} = \{a_{k,t}^i\}_{i \in \mathcal{N}}, k \in \{1, ..., K\}$, where $a^i$ is executed according to agent $i$'s policy $\pi^i : \mathcal{O}^i \to \Delta(\mathcal{A}^i)$, $K = |\mathcal{A}|$ is the size of action space. The environment then transitions to $s_{t+1}$ and rewards each agent $i$ by $r_t^i = R^i(s_t, \boldsymbol{a_t}, s_{t+1})$, where $R^i(\cdot)$ is the reward function. The goal of agent $i$ is to find the best policy $\pi^i$ that maximizes the total reward received from the environment from a start state to an end state. The expected cumulative discounted reward is expressed as $\mathbb{E}[\sum_{t=0}^{\infty} \gamma^t \cdot r_t^i]$. In this paper, we consider the $N$ agents, represented by $\boldsymbol{\pi} = \{\pi^1, ..., \pi^N\}$, with policies (actors) parameterized by $\boldsymbol{\theta} = \{\theta_1, ..., \theta_N\}$. Given a specific policy $\pi$, we adopt state value function the $V_{\gamma,\psi}^\pi$ to approximate expected return, where $\psi$ is the parameters of $V_{\gamma,\psi}^\pi$.

## 4    Problem Formulation

### 4.1    Reward Uncertainty in MARL

Reward uncertainty is prevalent in many real-world scenarios, but its influence is not well solved in the literature. Handling non-stationary reward during training is necessary for successfully learning complex behaviors in multi-agent environments. In many practical scenarios, the agents usually

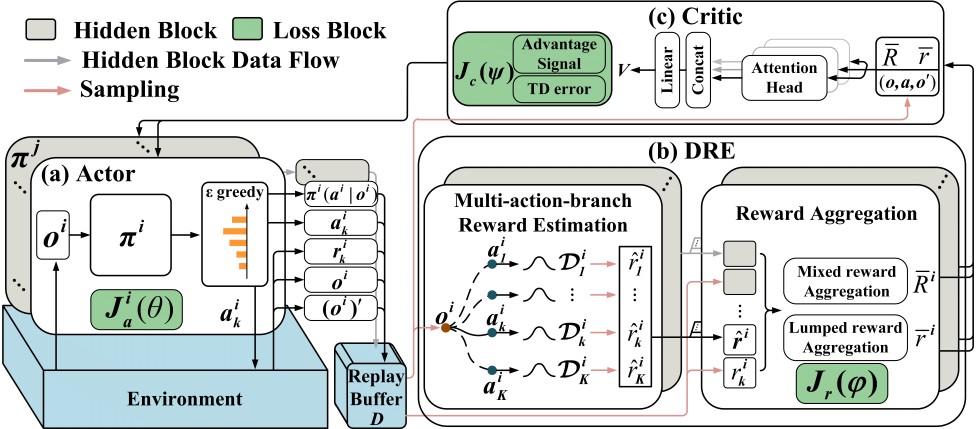

Figure 1: The overall architecture of Multi-agent Distributional Reward Estimation, which contains **(a) Decentralized Actors (b) Distributional Reward Estimation (c) Centralized Critic**. As is shown in **(a)**, we perform decentralized execution according to each agent's observation and store experience in the replay buffer. Then in **(b)**, we perform multi-action-branch reward estimation followed by policy-weighted reward aggregation. Finally, as shown in **(c)**, the centralized critic executes training with aggregated rewards and provides advantage signals for actors.

observe non-stationary reward feedback due to the interplay and inherent randomness rather than perfectly receiving the precise reward for training. Consider a multi-agent environment with high complexity and large state-action space such that it is intractable for agents to explore the entire state space. Suppose that we can only have access to the non-stationary reward. How can the agents learn cooperative behaviors under the reward uncertainty? We should figure out the factors that cause the reward uncertainty. Assume we are given a multi-agent scenario consisting of $N$ agents. the reward contains uncertainty that is caused by multiple factors that can be classified into two aspects.

**Mutual interaction.** The first factor that causes reward uncertainty is mutual interaction between agents. As mentioned in Section 1, the interplay between agents can lead to the one-to-many mapping between state and reward. For example, suppose that we need two agents to complete a task cooperatively. The agent will receive various rewards because its partner executes other actions even though the agent executes the same action under the same observation.

**Natural disturbance.** Another aspect lies in the natural disturbance of the environment. During the training process, the environment contains inherent randomness that is ubiquitous in the real world. For instance, the sensor's feedback fluctuates due to the variation in temperature and lighting, which makes the reward feedback inaccurate.

In the following section, we focus on the two aspects and propose multi-action-branch reward estimation and reward aggregation for providing stable updating signals for agents. Here, the intuition is that the decision should be executed after carefully considering all possible consequences. Finally, in Section 6, we empirically investigate the validity of our method in different scenarios.

### 4.2 Reward Estimation

Usually, many MARL scenarios provide precise human-designed rewards, which deviate from the actual situation in practice. To simulate the reward uncertainty, we define the stochastic rewards that are provided by the environment according to some stochastic processes. That is, the reward is generated from some distributions with certain probability densities [60]. To capture the reward uncertainty and facilitate training, reward estimation is proposed to reduce the impact of stochastic rewards [60, 38]. The reward estimator [40], which is a core component in a scenario with stochastic rewards, evaluates $s$, $(s, a)$, or $(s, a, s')$ pairs at time step $t$ and estimates the possible rewards that are used for guiding the agent through the user goal. Romoff et al. [38] model the task of learning the reward estimator as a simple point-to-point regression problem by function approximation: $\mathcal{L}(\varphi) = \mathbb{E}[(r - \tilde{R}(\mathcal{T}; \varphi))^2]$, where $\tilde{R}(\mathcal{T}; \varphi)$ is the reward estimator with parameter $\varphi$ based on different inputs $\mathcal{T} \in \{s, (s, a), (s, a, s')\}$.

# 5 DRE-MARL

In this section, we introduce a general training framework, Multi-agent Distributional Reward Estimation (DRE-MARL), for the reward uncertainty problem in MARL, as shown in Figure 1. We adopt the architecture of centralized training and decentralized execution (CTDE) [22], which consists of $N$ decentralized actors $\{\pi_\theta^i\}_{i \in \mathcal{N}}$ and a centralized critic $V_{\gamma,\psi}^\pi$, parameterized by $\theta$ and $\psi$, respectively. In practice, we utilize $\{\pi_\theta^i\}_{i \in \mathcal{N}}$ shown in Figure 1(a) to interact with the environment and collect experiences. Figure 1(b) shows the proposed distributional reward estimation structure, which consists of two stages: we first perform multi-action-branch reward distribution estimation from observed experiences and sample rewards from the reward distributions of action branches. Then we aggregate the environment rewards and sampled rewards to guide the training of the critic and actors. Figure 1(c) indicates that we utilize a graph attention network with multi-head attention [21, 4] to capture the global information from observations $(o^1, ..., o^N)$. The centralized critic can simultaneously produce advantage signals for all agents through one forward calculation during training. For stability, we construct target centralized critic $V_{\gamma,\tilde{\psi}}$[27, 43] and $N$ target policies $\tilde{\pi}_{\tilde{\theta}}$, parameterized by $\tilde{\psi}$ and $\tilde{\theta}$, respectively. The target network parameters $(\tilde{\psi}, \tilde{\theta})$ are only updated with current network parameters $(\psi, \theta)$ and are held fixed between individual updates [19, 54].

## 5.1 Multi-action-branch Reward Estimation

Modeling reward distributions is challenging because the reward uncertainty caused by agents' interplay grows exponentially with an increase of agent number. One direct method is to model reward distributions on joint state-action space. But this method suffers from huge uncertainty caused by mutual interaction as agent number grows. Besides, in the ablation study (See Figure 3 (left) for details.), we also verify that modeling joint reward distributions cannot resolve the issue.

To handle the above challenge, we propose another method to achieve our goal. We simplify the issue by viewing other agents as a part of the environment such that we only need to focus on the reward estimation of each agent instead of all agents. Additionally, inspired by the fact that humans will imagine the potential consequences based on historical experience, we equip each agent with a reward estimator $\widehat{R}^i$ to better capture the reward uncertainty and stabilize the training process. Specifically, for agent $i$ at time step $t$, we propose the multi-action-branch reward estimator $\widehat{R}^i(o_t^i, a_{k,t}^i; \varphi^i) \in \mathcal{D}$ to model the reward distribution based on agent $i$'s observation $o_t^i$ and $k$-th action $a_{k,t}^i$, where $\mathcal{D}$ represents reward distribution space, $\widehat{R}_k^i$ represents the reward distribution of agent $i$ in $k$-th action branch, and $\varphi^i$ is the parameters of $\widehat{R}^i$. We use $\widehat{R}^i \in \mathcal{D}^K$ to represent the $K$ estimated reward distributions for agent $i$. We can achieve the above goal by optimizing the overall objective function $J_r = \sum_i J_r^i(\varphi^i)$, and the loss function of every agent is as follows:

$$J_r(\varphi) = \mathbb{E}_{(o,a_k,r_k)\sim D}\left[-\log \mathbb{P}[r_k|\widehat{R}(o, a_k; \varphi)] + \mathcal{L}_{\widehat{R}}\right], \tag{1}$$

where we omit the superscript $i$ of $J_r(\varphi)$, $D$ is the replay buffer, and $-\log \mathbb{P}[\cdot|\cdot]$ is negative log likelihood. $\mathcal{L}_{\widehat{R}}$ is a regular term of reward distributions. In practice, we adopt Gaussian reward distribution $\mathcal{D}(\boldsymbol{\mu}, \boldsymbol{\sigma})$, but $\widehat{R}$ can be easily extended to other distributions. Then $\mathcal{L}_{\widehat{R}}$ can be defined by $\alpha \cdot \|\boldsymbol{\sigma}\|_1 + \beta \cdot var(\boldsymbol{\mu})$, where $var$ is the variance of $\boldsymbol{\mu}$. $\alpha$ and $\beta$ are hyperparameters.

Multi-action-branch reward estimation enables us to forecast possible rewards of all action branches. It is just like we only think about each specific situation separately, but in practice, we usually make a decision considering all possible consequences. Therefore, to evaluate the experience thoughtfully and stabilize training, we propose reward aggregation introduced in the following section to reduce the impact of reward uncertainty during the training process.

## 5.2 Training with Reward Aggregation

For each agent at each time step, the agent can only obtain a single reward $r_k$ while executing $k$-th action $a_k$. But we augment $r_k$ as a built-up reward vector, where the $k$-th of the vector is $r_k$ and the following parts are the estimated rewards $\hat{r}$ sampled from $\widehat{R}^i$. Then we aggregate the built-up reward vector with the policy-weighted operation. We mainly obtain two types of rewards after reward

aggregation: the mixed reward $\bar{R} \in \mathbb{R}$ and the lumped reward $\bar{r} \in \mathbb{R}$, where $\bar{R}$ is used to update centralized critic $V_{\gamma,\psi}$ and $\bar{r}$ is used to update decentralized actors $\{\pi_\theta^i\}_{i \in \mathcal{N}}$.

Specifically, for each agent we first define the built-up reward vector $m = h(\hat{\boldsymbol{r}}, r_k) \in \mathbb{R}^K$, which is constructed by replacing the $k$-th value of vector $\hat{\boldsymbol{r}}$ with true environment reward $r_k$ because the agent only takes $a_k$ during interaction with the environment, where $h(\cdot)$ is defined by $h(\hat{\boldsymbol{r}}, r_k) = [\hat{r}_1, ..., r_k, ..., \hat{r}_K]$. Then the mixed reward $\bar{R} = g(m^1, ..., m^N) \cdot \tilde{\pi}^i(\cdot|o)$ is calculated by policy-weighted aggregation, where $g(\cdot)$ represents two operations as below. 1) *MeanOperation* ($g_{MO}$): we utilize the average value of $m^1, ..., m^N$ as the output of $g(\cdot)$. 2) *SimpleSelection* ($g_{SS}$): we directly select $m^i$ for agent $i$ to calculate the mixed reward $\bar{R}$. Mathematically, $\bar{R}$ is defined by

$$\bar{R}^i = g(m^1, ..., m^N) \cdot \tilde{\pi}^i(\cdot|o) = \begin{cases} mean(m^1, ..., m^N) \cdot \tilde{\pi}^i(\cdot|o) & \text{if } g = g_{MO} \\ m^i \cdot \tilde{\pi}^i(\cdot|o) & \text{if } g = g_{SS} \end{cases}. \tag{2}$$

Sampling from the reward distributions has been proven to be beneficial for exploration in some settings [18, 60, 23]. But at the same time, it will also introduce more uncertainty during training. Here, we mainly focus on reducing the influence of reward uncertainty. Therefore we use the mean value of reward distributions to perform reward aggregation, usually achieving a more stable training process. These two schemes, the sampling option and mean option, can be chosen flexibly. It is a tradeoff: we choose the former option while the environment is hard to explore. The latter is more suitable for an environment with more uncertainty but is easy to explore.

Next, the centralized critic updates a parametric state value function $V_{\gamma,\psi}$ by minimizing the Bellman residual

$$J_c(\psi) = \mathbb{E}_{(o, r_k, o') \sim D} \left[ \bar{R} + \gamma V_{\gamma, \tilde{\psi}}(o') - V_{\gamma, \psi}(o) \right]^2, \tag{3}$$

where $\psi$ and $\tilde{\psi}$ are the parameters of current state value network and target state value network, respectively. Finally, for each agent during each training iteration, the decentralized actors follow the same training scheme of policy optimization with advantage function [41]:

$$J_a(\theta) = \mathbb{E} \left[ \min\left(u\left(\theta\right), \text{clip}\left(u\left(\theta\right), 1 - \epsilon, 1 + \epsilon\right)\right) \hat{A}_\gamma + \eta H(\pi_\theta) \right], \tag{4}$$

where $u(\theta) = \frac{\pi_\theta(a|o)}{\pi_{\tilde{\theta}}(a|o)}$ is the importance weight, $\epsilon$ is clip hyperparameter and usually we choose $\epsilon = 0.2$. $H(\pi_\theta^i)$ is the policy entropy of agent $i$, and $\eta$ controls the importance of entropy penalty objective. For agent $i$, the advantage value is defined as follows: $\hat{A}_\gamma^i = \bar{r}^i + \gamma V_{\gamma, \tilde{\psi}}(o') - V_{\gamma, \psi}(o)$, where $\bar{r}^i = l(m^1, ..., m^N, r_k)$ is the lumped reward that is defined by

$$\bar{r}^i = l(m^1, ..., m^N, r_k^i) = \begin{cases} mean(m^1, ..., m^N) & \text{if } l = l_{MO} \\ mean(m^i) & \text{if } l = l_{SMO} \\ r_k^i & \text{if } l = l_{SS} \end{cases}, \tag{5}$$

where $l$ represents three operations as follows. 1) *MeanOperation* ($l_{MO}$): we average on all agents' built-up rewards $m^i_{i \in \mathcal{N}}$ to obtain a single output. 2) *SimpleMeanOperation* ($l_{SMO}$): we utilize the mean value of agent $i$'s built-up rewards $m^i$ as output. 3) *SimpleSelection* ($l_{SS}$): we directly select the environment reward $r_k^i$ as the lumped reward.

Following the above procedure, we can update the reward estimators and policies iteratively. Further details and the full algorithm for optimizing DRE-MARL can be found in Appendix A.

## 6 Experiments

### 6.1 Environment Settings

To demonstrate the effectiveness of the proposed method, we provide experimental results in several benchmark MARL environments based on the multi-agent particle environments [22] (MPE) and several variants of MPE. Specifically, we consider cooperative navigation (CN), reference (REF), and treasure collection (TREA) environments from small to large agent numbers. In the following sections, we use CN-$q$, REF-$q$, and TREA-$q$ to represent different environment variants, where $q \in \{2, 3, 7, 10\}$ and the entire environment variants are {CN-3, CN-7, CN-10, REF-2, REF-7,

Table 1: Performance comparison of DRE-MARL and several SOTA MARL algorithms under the $r_{dete}$, $r_{dist}$, and $r_{ac-dist}$ settings. The values represent mean episodic rewards.

| | Scenario | q | DRE-MARL (ours) | p2p-MARL [38] | MAPPO [62] | MAAC [13] | QMIX [36] | MADDPG [22] | IQL [58] |
|---|---|---|---|---|---|---|---|---|---|
| $r_{dete}$ | CN-q | 3 | **-154**±11.90 | -167±12.80 | -425±99.47 | -227±43.38 | -1168±77.73 | -169±6.257 | -738±175.2 |
| | | 7 | **-3070**±89.18 | -3101±121.5 | -5674±801.5 | -3293±279.4 | -9320±1528. | -3428±45.11 | -5092±509.5 |
| | | 10 | **-8138**±285.6 | -8418±277.0 | -13375±1372. | -8555±873.0 | -21858±4320. | -8938±67.94 | -12658±864.4 |
| | REF-q | 2 | -54±8.589 | -58±8.638 | -85±39.74 | -69±28.30 | -438±205.0 | **-44**±2.394 | -95±17.50 |
| | | 7 | **-732**±73.85 | -808±88.99 | -1187±365.0 | -1120±302.5 | -1359±130.5 | -1133±50.92 | -2400±285.2 |
| | | 10 | **-1524**±140.9 | -1542±152.4 | -3317±882.3 | -2380±604.9 | -4037±235.3 | -3102±135.1 | -4921±449.2 |
| | TREA-q | 3 | -458±26.73 | -457±28.06 | -671±95.23 | -480±52.52 | **-390**±37.96 | -428±11.17 | -924±86.37 |
| | | 7 | **-2641**±134.6 | -2638±134.8 | -3648±0.005 | -2809±270.0 | -3324±270.9 | -2678±59.59 | -4733±246.7 |
| | | 10 | **-5076**±190.4 | -5260±280.8 | -6864±825.7 | **-4931**±438.1 | -8715±265.4 | -5699±150.8 | -8465±399.8 |
| $r_{dist}$ | CN-q | 3 | **-161**±13.74 | -186±13.73 | -773±103.9 | -270±55.01 | -1263±66.95 | -213±6.300 | -552±58.18 |
| | | 7 | **-3200**±93.43 | -3262±132.3 | -7481±1072. | -3558±419.7 | -10678±2076. | -3617±49.06 | -6141±1161. |
| | | 10 | **-8605**±277.0 | -8792±302.9 | -19696±2607. | -8986±902.7 | -22699±4151. | -9365±108.5 | -12693±2046. |
| | REF-q | 2 | -57±8.504 | -63±8.953 | -86±27.21 | -72±30.56 | -280±193.1 | **-43**±2.234 | -226±30.06 |
| | | 7 | **-752**±79.95 | -800±88.33 | -1947±639.3 | -1070±283.7 | -1894±195.3 | -1182±38.16 | -2514±283.6 |
| | | 10 | **-1676**±177.6 | **-1570**±156.0 | -5580±1745. | -2558±588.7 | -3584±338.6 | -2968±121.0 | -4090±1140. |
| | TREA-q | 3 | -484±25.97 | -447±26.12 | -691±136.3 | -513±76.72 | **-427**±39.78 | -449±8.016 | -889±49.29 |
| | | 7 | **-2733**±131.6 | -2745±150.0 | -3741±0.000 | -2956±313.2 | -3553±212.7 | -2824±75.14 | -5092±394.1 |
| | | 10 | **-5340**±223.6 | -5515±305.1 | -6912±876.3 | **-5306**±587.2 | -8897±285.3 | -5847±147.4 | -8727±464.4 |
| $r_{ac-dist}$ | CN-q | 3 | **477**±59.64 | 271±87.00 | 156±80.78 | -98±254.8 | 367±61.66 | 476±67.32 | -2645±740.2 |
| | | 7 | **888**±398.3 | -480±484.0 | -2960±1418. | -85±487.5 | -7122±1167. | 802±345.6 | -24492±4691. |
| | | 10 | **133**±610.0 | -2900±784.4 | -7379±1510. | -3036±1024. | -16977±4269. | -122±746.8 | -59231±8688. |
| | REF-q | 2 | **283**±26.21 | 213±32.72 | 247±46.78 | 106±18.96 | 255±23.77 | **287**±25.67 | -159±128.3 |
| | | 7 | **3323**±194.7 | 1886±269.5 | 2949±264.1 | 215±221.7 | 2824±214.2 | 2824±297.9 | -26±644.8 |
| | | 10 | **6340**±435.0 | 2884±575.9 | 5301±612.5 | -230±647.3 | 5229±453.1 | 4981±729.0 | 181±1331. |
| | TREA-q | 3 | 2567±123.9 | 1708±228.1 | **2742**±122.5 | 1776±124.7 | 2384±105.7 | 2712±107.7 | -3505±719.7 |
| | | 7 | 14249±547.1 | 9627±1154. | 14473±353.2 | 8342±463.5 | 14364±346.3 | **15436**±381.2 | -17986±2391. |
| | | 10 | 29193±1183. | 20437±2679. | 31878±850.5 | 17306±875.7 | 27100±1061. | **32421**±642.6 | -32136±5006. |
| | normalized mean performance | | **9.802** | 9.104 | 6.486 | 8.091 | 4.762 | 9.080 | 1.990 |

REF-10, TREA-3, TREA-7, TREA-10}. In all these environments, the reward is set to collaborative, which means the reward is the summation of individual rewards. Detailed experimental description and settings can be found in Appendix B.1.

**Reward formulation.** To simulate the reward uncertainty in these classic environments, we investigate several reward settings with induced uncertainty as follows. 1) *Deterministic reward* ($r_{dete}$): in classic environments, the reward is calculated with respect to distance [13, 22]. Suppose that the feedback is exactly precise. The deterministic reward refers to that the same observation-action pair is mapped to an identical reward. We use $r_{dete}^i$ to represent this reward setting. 2) *Natural disturbance reward* ($r_{dist}$): reward uncertainty caused by natural disturbance, mentioned in Section 4.1, can be regarded as the overall impact on the received reward because the sources of dynamics are coupled with each other. Similar to prior works [38, 60], we consider the natural disturbance reward that is designed by $r_{dist}^i \sim \mathcal{N}(r_{dete}^i, 1) * 0.05 + r_{dete}^i$. We use $r_{dist}^i$ to represent this reward setting. 3) *Mutual interaction reward* ($r_{ac-dist}$): in order to simulate the reward uncertainty caused by mutual interaction mentioned in Section 4.1, we add different reward distributions on different action branches such that the same observation-action pair will result in different rewards. Specifically, the action dist-reward is defined by $r_{ac-dist}^i \sim \mathcal{N}(u, \delta) + r_{dete}^i$, where $u = \text{Index}(a_k^i)$ and $\delta = 0.001$. We use $r_{ac-dist}^i$ to represent this reward setting. $r_{dist}$ and $r_{ac-dist}$ settings are particularly challenging because agents need to conquer more dynamics.

**Baselines.** In order to verify the efficacy of distributional reward estimation in MARL with reward uncertainty, we select the following methods as baselines: MADDPG [22] adopts a value-based centralized Q network to concatenate and process all agents' observations. MAAC [13] uses attention mechanism in CTDE paradigm. MAPPO [62] reaches good results with on-policy method. QMIX [36] adopts monotonicity to decompose team reward. IQL [58] adopts completely independent training on each agent. Apart from the above methods, we also compare with point-to-point (p2p-MARL) reward estimation [38] and global reward estimation (GRE-MARL).

**Metrics.** For all methods we consider, we report the final performance under different reward settings ($r_{dete}$, $r_{dist}$, $r_{ac-dist}$), which are introduced above. Following prior studies [13, 22], we set the episode

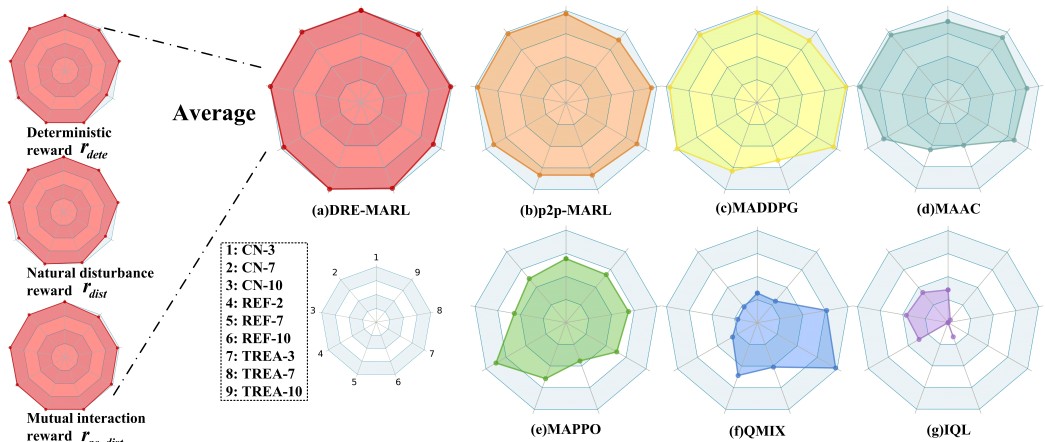

Figure 2: The normalized mean performance under different reward settings ($r_{dete}, r_{dist}, r_{ac-dist}$). Each polygon (shown right) is averaged on three sub-polygons (shown left), where each sub-polygon represents the corresponding reward setting. Each vertex of the sub-polygon denotes the normalized performance, which matches clockwise from CN-3 to TREA-10. The detailed calculation method of normalized performance can be found in Section 6.1.

length as 25 for all experiments and use the mean episode reward as the evaluation metric. Because the value of performance in different scenarios varies considerably, we also consider performance normalization based on all methods. For example, if we have performance values $(M_1, M_2, M_3)$ of three models, then the normalized performance is defined by $M'_v = \frac{\omega(M_v - min(M_1, M_2, M_3))}{max(M_1, M_2, M_3) - min(M_1, M_2, M_3)}$, where $\omega$ enables us to distinguish the differences of methods appropriately. We set $\omega = 10$ in the experiments. More details about training and implementation are provided in Appendix C.

## 6.2 Results and Analysis

Table 1 shows the performance of different methods with the three reward settings. We report the true value of each experiment. The results show that in most of the environments, DRE-MARL reaches the best performance, which reveals the effectiveness of distributional reward estimation. In several scenarios, such as TREA-7 with $r_{dete}$ and $r_{dist}$, DRE-MARL is not the best, but its performance is very close to the best. Under the $r_{ac-dist}$ reward uncertainty, DRE-MARL fails in TREA-3, TREA-7, and TREA-10. That is because $r_{ac-dist}$ brings more difficulties for attention-based methods in TREA: DRE-MARL and MAAC all adopt multi-head attention architecture, and they are subject to the same trend of changing. We find that in $r_{dete}$ and $r_{dist}$ settings, MAAC achieves the competitive performance, while our method reaches the best performance. But in the $r_{ac-dist}$ setting, the performance of MAAC is pretty bad, and also, DRE-MARL is not the best one. If we consider the utility of DRE based on attention in the setting of TREA with $r_{ac-dist}$ (i.e., Comparation between DRE-MARL and MAAC), DRE brings at least $50\%$ improvement.

To reveal the comprehensive abilities of the above methods, we normalize the performances of all methods and plot the overall performances as shown in Figure 2. The vertexes (1∼9) of the polygon represent the corresponding scenarios (CN-$q$,REF-$q$,TREA-$q$). To get the values on vertexes of polygon, we average the normalized values of performances under all reward settings $r_{dete}, r_{dist}, r_{ac-dist}$. Figure 2 suggests that DRE-MARL consistently achieves the best overall performance in almost all scenarios. In REF-2 and TREA-3, the performance of DRE-MARL is just slightly lower than MADDPG because the agent number is small, and it is relatively easy to train. Additionally, The results also illustrate that DRE-MARL has a more robust performance with respect to various reward uncertainties because the DRE can capture the reward uncertainty and stabilize training under different reward settings. Besides, exhaustive experiments are shown in Appendix B.

## 6.3 Ablation Study

Firstly, we perform the ablation study to investigate the impact of different reward estimation methods on the proposed model: 1)DRE-MARL: distributional reward estimation. 2) GRE-MARL:

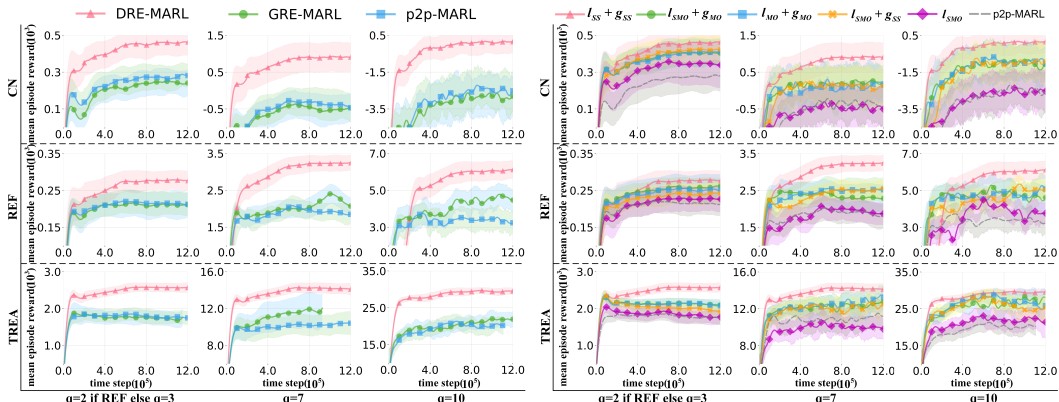

Figure 3: *Left*: Ablation study under reward setting $r_{ac-dist}$. We change the reward estimation strategy, such as global reward estimation (GRE-MARL) and point-to-point reward estimation (p2p-MARL), and probe the influence of different reward estimation methods. *Right*: Reward aggregation analysis on different aggregated schemes. We evaluate our model with different and frequently-used reward aggregation strategies, which enable us to assess our method comprehensively.

global reward estimation based on joint state-action pairs. 3) p2p-MARL: point-to-point reward estimation. As shown in Figure 3 (left), we report the learning curves of the above methods under the reward setting of $r_{ac-dist}$. The results show that DRE-MARL is able to achieve better asymptotic performance under different scenarios. GRE-MARL fails to capture the reward uncertainty because the joint state-action space of MARL is huge, which brings more difficulties for global reward estimation. p2p-MARL does not consider the reward uncertainty caused by mutual interactions among agents as the one-to-many mapping problem, so it performs badly by adopting the one-to-one reward estimation method in MARL scenarios.

To probe the effect of different reward aggregation schemes on the performance, we report the performance of several aggregation schemes in Figure 3 (right) under the $r_{ac-dist}$ setting. Recall the reward aggregation introduced in Section 5.2, we select the following aggregation schemes. 1) $l_{SS} + g_{SS}$, 2) $l_{SMO} + g_{MO}$, 3) $l_{MO} + g_{MO}$, 4) $l_{SMO} + g_{SS}$, 5) $l_{SMO}$ (no aggregation). In Figure 3 (right), we remark that the utility of reward aggregation, which makes the agent make decisions thoughtfully, consistently improves performance in various scenarios compared with GRE-MARL and p2p-MARL. Among all aggregation schemes, $l_{SS} + g_{SS}$ reaches the best performance, and the reported results are also based on this aggregation scheme. More details are shown in Appendix B.2.

## 7 Conclusion, Limitations, and Broader Impact

In this paper, we present DRE-MARL, a general and effective reward estimation method for reward uncertainty in MARL. To capture the reward uncertainty and stabilize the training process, we investigate the benefits of distributional reward estimation followed by reward aggregation in MARL when reward uncertainty is present. The intuition is that careful consideration of all possible consequences is useful for learning policies. For our proposed DRE-MARL, we propose two stages to perform distributional reward estimation: we first design the multi-action-branch reward estimation to model reward distributions on all action branches. Then we propose policy-weighted reward aggregation with environment rewards and sampled rewards. Here we emphasize that reward aggregation augments the reward from one action branch to all action branches and thus provides more precise updating signals for the critic and actors. Experimental results verify that agents learned by our framework can achieve the best performance and show strong robustness on different types of reward uncertainty. In conclusion, we hope that our work could demonstrate the potential of distributional reward estimation to improve the capacity of MARL when reward uncertainty is present and encourage future works to develop novel reward estimation methods.

In terms of limitations, DRE-MARL requires a prior assumption about the form of reward distribution. Just like tanglesome signals can be factorized into the superposition of simple but elementary sinusoidal signals, we may consider using a cluster of basic distributions as reward distribution in

future work. Another limitation is that reward aggregation is only used in discrete action space. Two solutions may be investigated in the future to solve this limitation: 1) Discretizing of the range of the available action value. 2) Learning a network with actions as inputs, just like how we convert discrete-action DQN [26] into continuous-action Q network in DDPG [19]. We do not see any negative societal impacts of our work while using our method in practice.

## Acknowledgments and Disclosure of Funding

We thank all anonymous reviewers for their constructive suggestions. This work is partially supported in part by the National Natural Science Foundation of China under grants Nos.61902145, 61976102, and U19A2065; the International Cooperation Project under grant No. 20220402009GH; and the National Key R&D Program of China under grants Nos. 2021ZD0112501 and 2021ZD0112502.

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
