# Appendix

## A  Pseudocode of DRE-MARL

The pseudocode for DRE-MARL training is shown in Algorithm 20, which takes the following steps. 1) We perform several interactions with the environment and collect experiences in advance. 2) As shown in lines $4-9$ (Algorithm 20), we collect transitions and deposit them in replay buffer $D$. 3) During the training process, i.e., lines $10-17$ (Algorithm 20), we update the centralized critic, decentralized actors, and reward estimators every 100 time steps following previous methods [13, 22] when the episode ends. 4) We will evaluate our model and refresh the replay buffer periodically. Source code is available at https://github.com/JF-Hu/DRE-MARL.git.

---

**Algorithm 1:** Multi-agent Distributional Reward Estimation (DRE-MARL)

---

**Input:** $N$ policies $\{\pi^i\}_{i\in\mathcal{N}}$ and target policies $\{\tilde{\pi}^i\}_{i\in\mathcal{N}}$ parameterized by $\boldsymbol{\theta} = \{\theta_i\}_{i\in\mathcal{N}}$ and $\tilde{\boldsymbol{\theta}} = \{\tilde{\theta}_i\}_{i\in\mathcal{N}}$, respectively; Centralized critic and target centralized critic parameterized by $\psi$ and $\tilde{\psi}$, respectively; $N$ reward estimators parameterized by $\boldsymbol{\varphi} = \{\varphi_i\}_{i\in\mathcal{N}}$; The environment with reward uncertainty

**Output:** $\psi, \boldsymbol{\theta}, \boldsymbol{\varphi}$

1 **Initialize:** $\psi, \tilde{\psi}, \boldsymbol{\theta}, \tilde{\boldsymbol{\theta}}, \boldsymbol{\varphi}$, reply buffer $D$
2 Pre-interact with environment
3 **for** *each iteration* **do**
4   **for** *each time step* $t$ **do**
5    Sample action $a_{k,t}^i \sim \pi^i(\cdot|o_t^i)$ for each agent $i$
6    Execute joint action $\boldsymbol{a_t}$ in the environment
7    Observe next observation $o_{t+1}^i$ and reward $r_{k,t}^i$ of each agent $i$
8    Store $\left(\{o_t^i\}, \{\pi^i(\cdot|o_t^i)\}, \{a_{k,t}^i\}, \{r_k^i\}, \{o_{t+1}^i\}\right)$ in replay buffer $D$
9   **end**
10   **if** *it's time to update* **then**
11    Sample a batch of transitions $\mathcal{B}$ from $D$
12    Update distributional reward estimators with Equation 1
13    Sampling rewards $\hat{\boldsymbol{r}}^{\boldsymbol{i}}$ from multi-action-branch reward distributions $\widehat{R}^i$
14    Perform reward aggregation with Equation 2 and Equation 5
15    Update centralized critic and decentralized actors
16    Update target networks $\{\pi_{\tilde{\theta}_i}\}_{i\in\mathcal{N}}, V_{\gamma,\tilde{\psi}}$: $\tilde{\theta}_i \leftarrow \tau\theta_i + (1-\tau)\tilde{\theta}_i, \tilde{\psi} \leftarrow \tau\psi + (1-\tau)\tilde{\psi}$
17   **end**
18   Evaluate individual policies periodically
19   Refresh replay buffer periodically
20 **end**

---

## B  Additional Experiments

### B.1  Environmental description

**Cooperation Navigation.** In this environment, three agents must collaborate to cover all landmarks and avoid colliding with each other simultaneously. The property of the received reward in this environment is set to be collaborative. For more complex environmental settings, we increase the number of agents and landmarks (i.e., Evaluation on 7 and 10 agents). We use the abbreviation CN to denote this environment.

**Reference.** It is a scenario with two agents and three landmarks. The difference between Cooperative Navigation and Reference is that the target landmark of each agent is only known to its partner. Thus every agent must convey correct information to each other to accomplish the task. To evaluate the capacity of our method in complicated environments, likewise the Cooperative Navigation, we also select 7 and 10 agents for evaluation. We use the abbreviation REF to denote this environment.

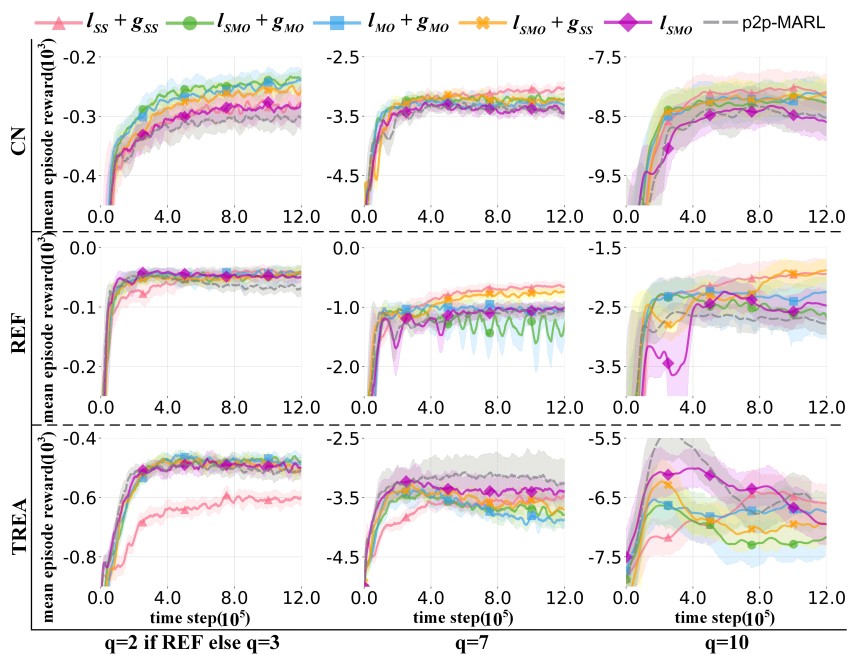

Figure 4: Comparison of the performance on different aggregated schemes while training with the $r_{ac-dist}$ setting and evaluating without the $r_{ac-dist}$ setting.

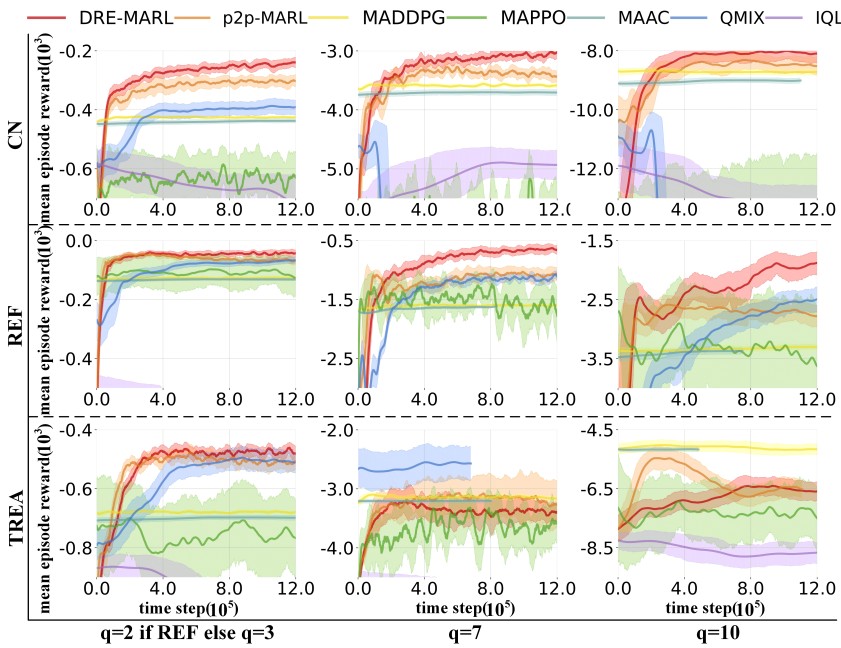

Figure 5: Comparison of DRE-MARL and different baselines while training with the $r_{ac-dist}$ setting and evaluating without the $r_{ac-dist}$ setting.

**Treasure Collection.** In this scenario, a successful process of treasure collection contains two stages: the first stage is that collectors cover the correct landmarks (i.e., collecting treasure), and the second is that the bank agents successfully receive the treasure. In this scenario, the per episode length is set to 25 because fewer steps bring more challenges to accomplish the two-stage task. The number of collectors and banks is the same, so the collectors should find their unique bank from all banks. Analogously, we choose 3c3b, 7c7b, 10c10b as measurements.

Table 2: Performance comparison of DRE-MARL, DRE-MARL variants, and several SOTA MARL algorithms while training with the individual rewards under the $r_{ac-dist}$ setting. The values represent mean episodic rewards.

| Reward Setting | $r_{ac-dist}$ | | | | | | |
|---|---|---|---|---|---|---|---|
| Scenario | CN-$q$ | | | REF-$q$ | | | Normalized Performance |
| $q$ | 3 | 7 | 10 | 2 | 7 | 10 | |
| DRE-MARL ($l_{SS} + g_{SS}$) | $129_{\pm26.93}$ | $36_{\pm77.89}$ | $-122_{\pm113.8}$ | $23_{\pm64.11}$ | $-13_{\pm201.3}$ | $402_{\pm108.3}$ | 7.893 |
| DRE-MARL ($l_{SMO} + g_{MO}$) | $126_{\pm26.54}$ | $11_{\pm75.25}$ | $-172_{\pm116.4}$ | $82_{\pm40.48}$ | $258_{\pm68.25}$ | $278_{\pm83.82}$ | 8.467 |
| DRE-MARL ($l_{MO} + g_{MO}$) | $144_{\pm22.80}$ | $43_{\pm66.72}$ | $-163_{\pm78.67}$ | $113_{\pm17.16}$ | $310_{\pm55.84}$ | $454_{\pm72.59}$ | 9.360 |
| DRE-MARL ($l_{SMO} + g_{SS}$) | $133_{\pm22.79}$ | $39_{\pm70.88}$ | $-86_{\pm113.8}$ | $11_{\pm61.19}$ | $177_{\pm120.8}$ | $74_{\pm133.6}$ | 7.904 |
| DRE-MARL ($l_{SMO}$) | $99_{\pm24.99}$ | $1_{\pm69.99}$ | $-186_{\pm120.6}$ | $44_{\pm45.23}$ | $171_{\pm87.49}$ | $300_{\pm89.35}$ | 7.737 |
| p2p-MARL | $92_{\pm25.27}$ | $-15_{\pm72.02}$ | $-171_{\pm116.9}$ | $35_{\pm50.32}$ | $192_{\pm101.8}$ | $227_{\pm87.09}$ | 7.517 |
| GRE-MARL | $113_{\pm26.66}$ | $-18_{\pm60.79}$ | $-249_{\pm98.28}$ | $116_{\pm16.30}$ | $244_{\pm67.66}$ | $231_{\pm98.08}$ | 8.171 |
| MADDPG | $138_{\pm30.59}$ | $47_{\pm84.41}$ | $-217_{\pm125.42}$ | $-108_{\pm102.8}$ | $-371_{\pm166.0}$ | $-609_{\pm196.7}$ | 4.608 |
| MAPPO | $10_{\pm43.89}$ | $-313_{\pm168.2}$ | $-686_{\pm299.0}$ | $-50_{\pm89.86}$ | $-103_{\pm243.2}$ | $-61_{\pm312.4}$ | 2.457 |
| MAAC | $132_{\pm17.02}$ | $34_{\pm47.49}$ | $-146_{\pm70.89}$ | $11_{\pm25.14}$ | $26_{\pm53.52}$ | $31_{\pm66.36}$ | 7.324 |
| QMIX | $-4_{\pm70.57}$ | $-209_{\pm170.6}$ | $-715_{\pm195.83}$ | $125_{\pm15.78}$ | $380_{\pm42.38}$ | $577_{\pm45.12}$ | 5.970 |
| IQL | $-66_{\pm42.65}$ | $-309_{\pm115.3}$ | $-705_{\pm151.5}$ | $-62_{\pm76.95}$ | $-210_{\pm137.1}$ | $-271_{\pm173.8}$ | 1.204 |

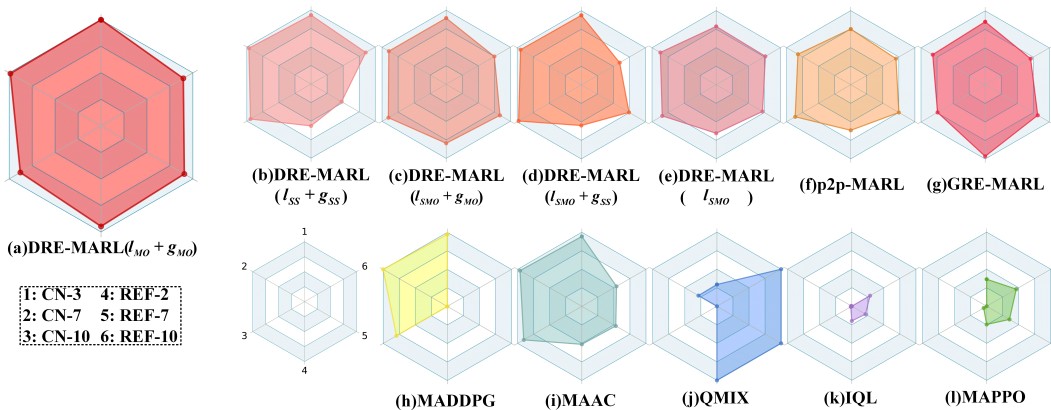

(a)DRE-MARL($l_{MO} + g_{MO}$)

(b)DRE-MARL ($l_{SS} + g_{SS}$)  (c)DRE-MARL ($l_{SMO} + g_{MO}$)  (d)DRE-MARL ($l_{SMO} + g_{SS}$)  (e)DRE-MARL ( $l_{SMO}$ )  (f)p2p-MARL  (g)GRE-MARL

1: CN-3   4: REF-2
2: CN-7   5: REF-7
3: CN-10  6: REF-10

(h)MADDPG  (i)MAAC  (j)QMIX  (k)IQL  (l)MAPPO

Figure 6: The normalized performance while training with the individual rewards and evaluating with the $r_{ac-dist}$ setting.

## B.2 Additional aggregation analysis

There are two choices when evaluating the proposed method. 1) Evaluating in the environment with the reward uncertainty setting as same as training. 2) Evaluating the environment without the setting of reward uncertainty. We usually select the first choice in practice because the training and testing environments are the same. The second choice is also important because it can show whether our model is stuck in the setting of reward uncertainty. We have report the results of first choice in Table 1, Figure 2, and Figure 3. In the next parts, we report the performance of different reward aggregation schemes and other baselines based on the second choice.

We evaluate the performance under the team reward setting in the environments where the $r_{ac-dist}$ is not added. As shown in Figure 4, in most environments, the proposed reward aggregation schemes achieve better performance than p2p-MARL. The aggregation scheme $l_{SMO} + g_{SS}$ is better than $l_{SS} + g_{SS}$ while Figure 3 (right) shows that $l_{SS} + g_{SS}$ is the best. This enlightens us different aggregation schemes are suitable for various purposes. In other words, if we want to train a model in precise environments designed by humans, we should choose $l_{SMO} + g_{SS}$ rather than $l_{SS} + g_{SS}$.

Table 3: Performance comparison of DRE-MARL, DRE-MARL variants, and several SOTA MARL algorithms while training with the individual rewards and evaluating without the $r_{ac-dist}$ setting. The values represent mean episodic rewards.

| Reward Setting | $r_{ac-dist}$ | | | | | | |
|---|---|---|---|---|---|---|---|
| Scenario | CN-$q$ | | | REF-$q$ | | | Normalized Performance |
| $q$ | 3 | 7 | 10 | 2 | 7 | 10 | |
| DRE-MARL ($l_{SS} + g_{SS}$) | -403$\pm$33.82 | -4148$\pm$416.8 | -9283$\pm$442.9 | -358$\pm$112.2 | -5663$\pm$2102. | -4711$\pm$996.3 | 7.016 |
| DRE-MARL ($l_{SMO} + g_{MO}$) | -320$\pm$32.84 | -3904$\pm$204.1 | -9775$\pm$396.8 | -166$\pm$81.60 | -2183$\pm$300.2 | -5744$\pm$668.7 | 8.457 |
| DRE-MARL ($l_{MO} + g_{MO}$) | -230$\pm$15.54 | -3260$\pm$126.8 | -8491$\pm$223.9 | -70$\pm$13.92 | -1361$\pm$175.7 | -2882$\pm$384.3 | 9.797 |
| DRE-MARL ($l_{SMO} + g_{SS}$) | -283$\pm$23.55 | -3587$\pm$158.7 | -8791$\pm$388.5 | -310$\pm$113.5 | -2825$\pm$800.7 | -7847$\pm$1756. | 7.920 |
| DRE-MARL ($l_{SMO}$) | -350$\pm$22.56 | -3672$\pm$161.9 | -9543$\pm$428.2 | -195$\pm$49.99 | -2161$\pm$382.7 | -4421$\pm$536.6 | 8.574 |
| p2p-MARL | -367$\pm$23.72 | -3717$\pm$160.7 | -9241$\pm$397.3 | -212$\pm$51.14 | -2213$\pm$452.2 | -4315$\pm$489.4 | 8.591 |
| GRE-MARL | -467$\pm$29.11 | -4522$\pm$165.5 | -11408$\pm$351.5 | -103$\pm$16.75 | -2604$\pm$435.7 | -6964$\pm$773.8 | 7.715 |
| MADDPG | -429$\pm$6.242 | -3589$\pm$42.26 | -8693$\pm$66.01 | -133$\pm$3.568 | -1635$\pm$21.71 | -3348$\pm$35.76 | 8.986 |
| MAPPO | -1074$\pm$240.4 | -9349$\pm$2315. | -21475$\pm$5276. | -603$\pm$216.0 | -6357$\pm$2021. | -11532$\pm$3721. | 1.007 |
| MAAC | -441$\pm$5.511 | -3732$\pm$35.19 | -9076$\pm$77.81 | -140$\pm$2.933 | -1712$\pm$23.21 | -3504$\pm$40.08 | 8.817 |
| QMIX | -671$\pm$234.8 | -4870$\pm$1128. | -14964$\pm$3096. | -66$\pm$10.64 | -1128$\pm$100.0 | -2327$\pm$145.4 | 7.790 |
| IQL | -973$\pm$38.33 | -6053$\pm$918.8 | -13836$\pm$1331. | -631$\pm$75.07 | -7905$\pm$381.7 | -16048$\pm$744.0 | 2.081 |

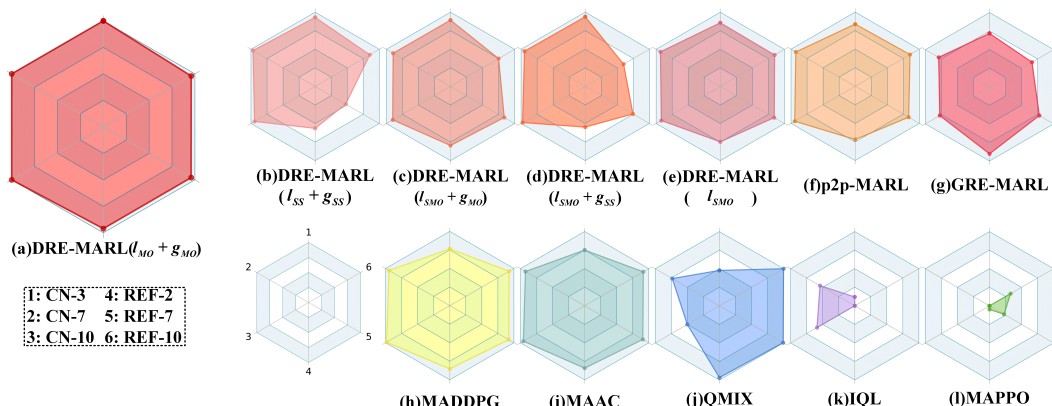

(a)DRE-MARL($l_{MO}+g_{MO}$)

(b)DRE-MARL ($l_{SS}+g_{SS}$)   (c)DRE-MARL ($l_{SMO}+g_{MO}$)   (d)DRE-MARL ($l_{SMO}+g_{SS}$)   (e)DRE-MARL ($l_{SMO}$)   (f)p2p-MARL   (g)GRE-MARL

(h)MADDPG   (i)MAAC   (j)QMIX   (k)IQL   (l)MAPPO

1: CN-3   4: REF-2
2: CN-7   5: REF-7
3: CN-10   6: REF-10

Figure 7: The normalized performance while training with the individual rewards and evaluating without the $r_{ac-dist}$ setting.

But if we are given an environment with reward uncertainty and the evaluation is also performed in it, the aggregation scheme $l_{SS} + g_{SS}$ is a better choice.

Furthermore, we also evaluate the performance while training with individual rewards, i.e., the agents can only receive the individual rewards rather than the team rewards during the training process. This environment setting is more complicated than the team reward setting because the agents can cause invalid updating on the centralized critic. The reward aggregation can provide a consistent updating direction for the centralized critic, so as illustrated in Table 2 and Table 3, our method achieves a better performance than the other baselines. The aggregation scheme $l_{MO} + g_{MO}$ reaches the best performance because the mean operation offers a relatively consistent direction of updating.

## B.3 Additional experiment results

In order to verify whether our model will be stuck in the setting of reward uncertainty, we evaluate the proposed method and various baselines in the environments without reward uncertainty. We report the learning curves in Figure 5. DRE-MARL performs better than different baselines, which illustrates that DRE-MARL can reduce the impact of reward uncertainty by adopting multi-action-branch reward estimation and policy-weighted reward aggregation during the training process.

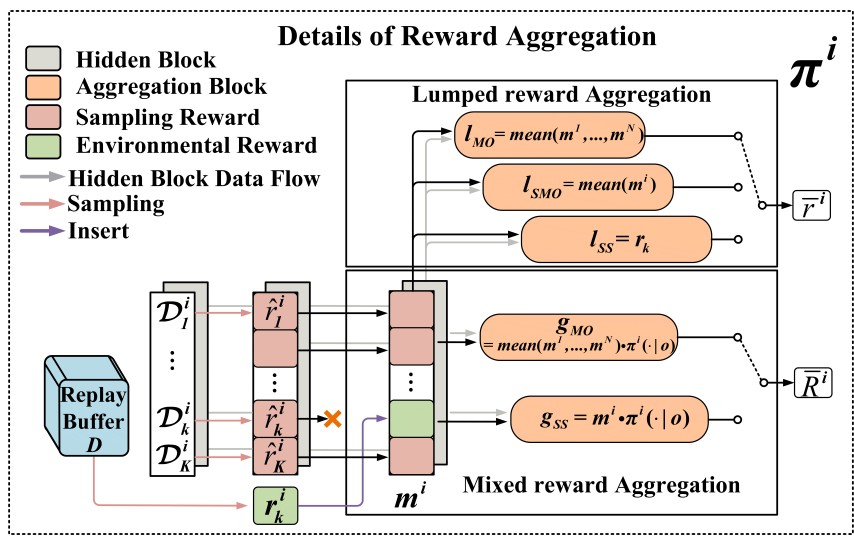

Figure 8: Graphical description of reward aggregation.

Multi-action-branch reward estimation followed by reward aggregation not only provides a solution to reduce reward uncertainty by augmenting reward from one action branch to all action branches but also offers a good way to improve collaboration by considering the aggregated rewards of all agents. In order to evaluate the performance just given the individual rewards, we set the environment to be non-collaborative during the training process, i.e., the agents can only receive their individual rewards rather than the team reward. Table 2 and Table 3 show the performance of baselines and different aggregation schemes in several scenario variants. The comprehensive abilities of the above methods are shown in Figure 6 and Figure 7. DRE-MARL ($l_{MO} + g_{MO}$) achieves better performance than other aggregation schemes and baselines in most environments. The reason is that reward aggregation provides a good solution to consider the holistic impact of individual agents when updating the centralized critic so as to achieve the common goal. Although QMIX reaches higher performance in REF, the difference is slight, and the overall performance of DRE-MARL ($l_{MO} + g_{MO}$) is obviously better than QMIX. Besides, in order to train the QMIX model, we actually calculate the team reward for QMIX during training, which violates the environment setting of training with the individual rewards. The value decomposition adopted by QMIX can be regarded as one backward method from the team reward to the individual rewards for solving MARL tasks. Our method provides a brand-new forward solution from the individual rewards to the team reward by using multi-action-branch reward estimation followed by policy-weighted reward aggregation.

## C   Implement Details

**Details of Reward Aggregation.**     For each agent, we first sample rewards $\hat{r}^i$ from estimated reward distributions $\mathcal{D}^i$. Then, we construct the built-up reward vector $m^i$ by replacing the value of $\hat{r}^i$ on the $k$-th action branch with the environment reward $r^i_k$. Next, we aggregate $\{m^i\}_{i \in \mathcal{N}}$ with different operations, as shown in Figure 8. Finally, we obtain the mixed reward $\bar{R}$ and the lumped reward $\bar{r}$ to train the centralized critic and the decentralized actors, respectively. The details of reward aggregation are shown in Figure 8.

**Network Architecture.**   The decentralized actors and distributional reward estimation networks adopt the simple fully-connected feedforward neural network with three layers in our framework. The two hidden layers' units are 64. $|\mathcal{A}|$ and $2|\mathcal{A}|$ are the number of output units in actors and distributional reward estimation networks, respectively, where $|\mathcal{A}|$ represents the size of action space. The centralized critic uses a graph attention neural network with eight attention heads, and each head's hidden unit is set to 8 to capture the dynamic relationship between agents. After the graph attention neural network, a two-hidden-layer fully-connected feedforward neural network is used to get the state value. Besides, we choose leaky relu as nonlinear activation for all networks.

**Compute.** Experiments are carried out on NVIDIA GeForce RTX 1080Ti GPUs and with fixed hyperparameter settings, which are described in the following. Each run of the experiments spanned about 2-12 hours, depending on the algorithm and the agents' number in the environment.

**Hyperparameters.** For all scenarios, the per-episode length is set to 25. The periodical replay buffer learning rate is 0.4, the learning interval is set to 4 episodes, and the entropy scale is set to 0.3. Detailed hyperparameters can be found in Table 4.

Table 4: The hyperparameters of DRE-MARL.

| Hyperparameter | Value |
|---|---|
| optimizer | Adam [14] |
| learning rate | $1 \cdot 10^{-3}$ |
| entropy scale | 0.1/0.3 |
| number of hidden layers (all networks) | 2 |
| nonlinearity of hidden layers (all networks) | Leaky ReLU |
| number of hidden units per layer (all networks) | 64 |
| number of attention head | 8 |
| time difference (TD) | 1 |
| buffer clear rate | 0.4 |
| discount ($\gamma$) | 0.95 |
| batch size | 1024 |
| $\varepsilon$-greedy | 0.7→0.9 |
| tau ($\tau$) | 0.01 |
| $\alpha$ | 0.1 |
| $\beta$ | 10 |
| $\eta$ | 0.3 |

## D More discussion of limitations

**Continuous action space.** In our method, the number of action branches matches the number of available discrete actions, so we can set the value of K to be equal to the number of available discrete actions such as "move forward", "move backward", "move left", "move right", and "motionless" in MPE. But in continuous action space, for example, we want to manipulate a robot arm. The "grasping" action needs us to assign a continuous value such as rotation angle to the robot arm. Right now, the available action is a range, so we can not define the number of K. We discuss the following possible solutions to solve the problem of continuous action space.

- One possible solution may be the discretization of the range of the available action value. More sophisticated discretization will bring better manipulation, but it will consume more computational resources at the same time. Although coarse discretization reduces the consumption of physical time, it may hurt the performance.

- Another possible solution is learning a network with actions as inputs, just like how we convert discrete-action DQN into continuous-action Q network in DDPG. The implicit reward function makes it possible to perform reward aggregation by sampling several discrete action points.

**Prior Gaussian distribution.** The reward distribution is indeed a bit hard to select in practice. However, just like tanglesome signals can be factorized into the superposition of simple but elementary sinusoidal signals, we may consider using a cluster of basic distributions as reward distribution in future work.

Table 5: The comparison of the computational cost about different models based on the CN-3 scenario. The items of the comparison contain three aspects: parameters, physical training time (min), and max memory consumption (MB).

| comparison items | | model | | | | | |
|---|---|---|---|---|---|---|---|
| | | DRE-MARL | MAPPO | MAAC | QMIX | MADDPG | IQL |
| para-meters | total | 198440 | 72854 | 450364 | 307330 | 172884 | 564642 |
| | trainable | 54126 | 72854 | 450364 | 123488 | 86436 | 141147 |
| | untrainable | 144314 | 0 | 0 | 183842 | 86448 | 423495 |
| physical training time (min) | | $256.0_{\pm 1.859}$ | $336.0_{\pm 9.623}$ | $915.9_{\pm 6.270}$ | $867.1_{\pm 16.54}$ | $166.9_{\pm 1.976}$ | $217.4_{\pm 2.311}$ |
| max memory consumption (MB) | | $1354._{\pm 60.18}$ | $310.6_{\pm 0.297}$ | $229590.1_{\pm 65.69}$ | $2146.5_{\pm 22.95}$ | $3754._{\pm 227.3}$ | $668.4_{\pm 20.18}$ |

Table 6: Performance comparison of DRE-MARL with and without regularization term $L_R$ while training with the team rewards and evaluating without the $r_{ac\text{-}dist}$ setting. The values represent mean episodic rewards.

| comparison items | | model |
|---|---|---|
| | | DRE-MARL |
| $l_{SS} + g_{SS}$ | with $L_R$ | $-272.7_{\pm 25.22}$ |
| | without $L_R$ | $-274.6_{\pm 19.13}$ |
| $l_{SMO} + g_{SS}$ | with $L_R$ | $-235.1_{\pm 16.38}$ |
| | without $L_R$ | $-365.0_{\pm 25.19}$ |
| $l_{MO} + g_{MO}$ | with $L_R$ | $-242.1_{\pm 16.15}$ |
| | without $L_R$ | $-258.9_{\pm 17.36}$ |
| $l_{MO} + g_{SS}$ | with $L_R$ | $-252.6_{\pm 17.49}$ |
| | without $L_R$ | $-354.2_{\pm 23.47}$ |
| no reward estimation | | $-258.2_{\pm 20.75}$ |

Table 7: Performance comparison of DRE-MARL variants and DFAC variants. We conduct the experiment in CN-3 with DFAC variants which is based on SC II originally, and adopt the original hyperparameters of DFAC variants. The DFAC-diql(128) denotes the number of hidden neural is 128.

| model | performance |
|---|---|
| $l_{SS} + g_{SS}$ | $-272.7_{\pm 25.22}$ |
| $l_{SMO} + g_{SS}$ | $-235.1_{\pm 16.38}$ |
| $l_{MO} + g_{MO}$ | $-242.1_{\pm 16.15}$ |
| $l_{MO} + g_{SS}$ | $-252.6_{\pm 17.49}$ |
| DFAC-diql(128) | $-786.9_{\pm 180.2}$ |
| DFAC-diql(256) | $-1117.9_{\pm 21.38}$ |
| DFAC-dmix(128) | $-1121.6_{\pm 21.51}$ |