# OpenReview forum: "Distributional Reward Estimation for Effective Multi-agent Deep Reinforcement Learning"
_NeurIPS.cc/2022/Conference — NeurIPS 2022 Accept_

### Official Review · Reviewer_ugkM · 2022-07-02

**Rating:** 5
**Confidence:** 2
**Soundness:** 3 good
**Presentation:** 3 good
**Contribution:** 3 good

**Summary:**

this paper propose distributional reward estimation for multi-agent reinforcement learning (DRE-MARL). this paper focuses on the problem of reward uncertainty in MARL. The main idea of this paper is to design the multi-action-branch  reward estimation and policy-weighted reward aggregation for stabilized training. This former part is simply function approximation with historical data, while the latter part is weighted reward aggregation. Experiments show that DRE-MARL outperforms other SoTA algorithms comprehensively.

**Questions:**

I suggest the authors put the related work in section 1 introduction to section 2.

I think the data volume for reward estimator should be very large, so I suggest the authors conducting some experiments to test how much data should be used.

**Limitations:**

the reward distribution in reward estimation is hard to choose

**Strengths And Weaknesses:**

Strengths:

it is novel to solve the reward uncertainty problem in MARL

policy-weighted reward aggregation enables stable training of the critic and actors and it is quite robust.

---

> ### Author Response · Authors · 2022-08-02
> **The response to Reviewer ugkM.**
>
> We are particularly encouraged that the reviewer finds our method novel and effective. We appreciate the valuable feedback of Reviewer ugkM and respond to the questions and limitations below.
>
>
> ### [I]. Explanation of questions
>
> > **[1/2] Q1:** I suggest the authors put the related work in section 1 introduction to section 2.
>
> For Q1, following your suggestions and at the same time preserving logical integrity, we revise the related work of Section 1 Introduction and Section 2 Related Work.
>
>
> > **[2/2] Q2:** I think the data volume for reward estimator should be very large, so I suggest the authors conducting some experiments to test how much data should be used.
>
>
> We appreciate your comments on the data volume problem. **We clarify that the data volume used to train reward estimators is not very large**. We do not need to collect extra data from the environment. The total data volume is identical to the baselines. In each training epoch, we train each agent's reward network the same as actor and critic, and the quantities of samples are also the same. We update our model every 100 timesteps with the batch size 1024. Additionally, we report the comparison of the training time (min) in Table 1, which illustrates the training cost is not expensive.
>
> **Table 1:** The comparison of the computational cost of different models based on the CN-3 scenario. The items of the comparison contain three aspects: parameters, physical training time (min), and max memory consumption (MB).
> | comparison items | DRE-MARL (ours) | MAPPO | MAAC | QMIX | MADDPG | IQL
> | -------- | -------- | -------- |-------- | -------- |-------- | -------- |
> | total  parameters   | 198440 | 72854 | 450364| 307330|172884 |564642 |
> | trainable  parameters | 54126   | 72854  | 450364 | 123488 | 86436 | 141147 |
> | untrainable  parameters | 144314   | 0  | 0 | 183842 | 86448 | 423495 |
> | physical training time (min) | 256.0±1.859   | 336.0±9.623  | 915.9±6.270 | 867.1±16.54 | 166.9±1.976 | 217.4±2.311 |
> | max memory consumption (MB) | 1354.±60.18   | 310.6±9.623  | 229590.1±65.69 | 2146.5±22.95 | 3754.±227.3 | 668.4±20.18 |
>
>
>
>
>
> ### [II]. Explanation of limitations
>
> > **[1/1] L1:** The reward distribution in reward estimation is hard to choose
>
>
> For limitations, we have discussed them in Section 7. Gaussian distribution is one of the most widely used distributions because of its powerful fitting ability, so we choose it as the form of reward distribution. We admit that choosing the reward distribution is indeed a bit hard in practice. However, just like tanglesome signals can be factorized into the superposition of simple but elementary sinusoidal signals, we may consider using a cluster of basic distributions as reward distribution in future work.
>
> ---
>
> We again thank Reviewer ugkM for reviewing our paper and giving suggestions. We hope our answers have addressed all the concerns the Reviewer has. If so, we would greatly appreciate it if Reviewer ugkM could consider raising their score. Please let us know if there are more questions.
>
> Paper2649 Authors

---

### Official Review · Reviewer_oHz3 · 2022-07-06

**Rating:** 7
**Confidence:** 4
**Soundness:** 3 good
**Presentation:** 4 excellent
**Contribution:** 3 good

**Summary:**

The paper outlines a new method for multi-agent reinforcement learning settings (DRE-MARL) which primarily focuses on reward estimation for multi-agent reinforcement learning settings. The authors first describe the problem settings and motivate why reward estimation is important and challenging in multi-agent reinforcement learning and how DRE-MARL differs from prior approaches, including reward uncertainty method and reward estimation methods in single-agent reinforcement learning. According to the authors' summary, many reward estimation methods struggle in multi-agent setting due to the additional complexity and greater sources of uncertainty. The authors identify mutual interaction between agents and natural disturbance of the environment as two sources of uncertainty in MARL settings and propose a method simulate reward uncertainty by applying stochastic processes.

Subsequently the authors outline their method (DRE-MARL) for reward estimation based on a distribution of potential action a single agent can take in a MARL setting and also describe different reward aggregation methods that are tested in the experiments. The authors then compare their method against different MARL baselines across a set of collaborative MARL environments showing mostly outperformance and conduct an ablation study for various reward aggregation schemes.

**Questions:**

**Nits**
- A is not capitalized in title "Multi-agent"
- first page says 35th Neurips, which is 2021
- Line 173-175 (page 5) have some spelling errors ("interplay grows exponentially with an increase of the agent number")

**General Questions:**
- Could you clarify why you choose a centralized critic for your method? Is this related to the environment setting (i.e. the environment only provides a single reward for all agents)? Can you see your method working with a de-centralized critic? what would change if anything?
- Could you say more about how your method might perform in competitive MARL settings? Right now you have looked at cooperative settings and it would be interesting to contrast that with competitive settings. (No new experiments needed, mainly looking for additional detail)
- It seems like p2p-MARL is the most competitive method to DRE-MARL and also uses reward estimation. Could you clarify the differences between p2p-MARL and DRE-MARL?
- How did you choose k for the distributional estimation? Was there a significant difference between different values of k?

**Limitations:**

I think that the paper could be improved by further discussion of the limitations of DRE-MARL. As far as I can tell only one limitation is briefly mentioned on Page 9, which is that DRE-MARL and reward aggregation are only used in discrete action spaces. It would be good to get further clarity on the following from the authors:
- How could DRE-MARL be applied to continuous action spaces?
- What limitations does DRE-MARL have that can inspire future work?

**Strengths And Weaknesses:**

**Originality:**
- Strengths: The paper proposes a new method for reward estimation in MARL settings that differs from prior approaches and shows good performance compared to baseline methods. Relevant work is cited and compared against in the beginning of the paper.
- Weaknesses: The authors could have provided more detail on the limitations of their method (discussed below as well)

**Quality:**
- Strengths: The methods and contributions are generally well supported in the experiments and discussed in detail in the paper.

**Clarity:**
- Strengths: The paper is generally well written and well organized with relevant equations, diagrams and descriptions.
- Weaknesses: The authors could have made better use of figure captions (in Figure 2 and Figure 3 specifically) to make it easier for the reader to understand the relevant messages conveyed by the figures.

**Significance:**
- Strengths: The paper proposes a new method in a relevant subject area of MARL.
- Weaknesses: The authors could have provided more detail on the limitations of their method and put it into the broader context of MARL methods (discussed below as well).

---

> ### Author Response · Authors · 2022-08-02
> **The response to Reviewer oHz3.**
>
> We are particularly encouraged that the reviewer finds our method novel and effective. We appreciate the valuable feedback of Reviewer oHz3 and respond to the weaknesses, questions, and limitations below.
>
> ### [I]. Explanation of strengths and weaknesses
>
> > **[1/3] W1:** The authors could have made better use of figure captions (in Figure 2 and Figure 3 specifically) to make it easier for the reader to understand the relevant messages conveyed by the figures.
>
>
> We really appreciate your comments, and we will add more explanations in the captions of Figure 2 and Figure 3 to make it easy for the readers. For more details, please refer to Figure 2 and Figure 3 in the main context.
>
>
> > **[2/3] W2:** The authors could have provided more detail on the limitations of their method (discussed below as well)
>
>
> We merge the responses to **W2**, **W3**, and limitations, and we put them in part of **III.L1**.
>
>
> > **[3/3] W3:** The authors could have provided more detail on the limitations of their method and put it into the broader context of MARL methods (discussed below as well).
>
>
> We merge the responses to **W2**, **W3**, and limitations, and we put them in part of **III.L1**.
>
>
>
> ### [II]. Explanation of questions
>
> > **[1/5] Q1:**
> > - A is not capitalized in title "Multi-agent"
> > - first page says 35th Neurips, which is 2021
> > - Line 173-175 (page 5) have some spelling errors ("interplay grows exponentially with an increase of the agent number")
>
>
> According to your suggestions, all the typos are revised. Thank you very much for pointing them out.
>
>
> > **[2/5] Q2:** Could you clarify why you choose a centralized critic for your method? Is this related to the environment setting (i.e. the environment only provides a single reward for all agents)? Can you see your method working with a de-centralized critic? what would change if anything?
>
>
> For Q2, centralized critic is commonly used as in MADDPG and MAAC. Our algorithm can be combined with decentralized critic. For example, it is feasible to design decentralized critic if there are only two agents in certain environments. However, when the number of agents is relatively large, equipping every agent with a critic brings a lot more computational cost and possible instability in training.
>
>
> > **[3/5] Q3:** Could you say more about how your method might perform in competitive MARL settings? Right now you have looked at cooperative settings and it would be interesting to contrast that with competitive settings. (No new experiments needed, mainly looking for additional detail)
>
>
> For Q3, although we only test the performance in cooperative settings, our method can also be used in competitive environments. When we put this method in competitive scenarios, we should figure out that the reward uncertainty comes not only from the teammate-agent group's mutual interaction and natural disturbance but also from the opposite-agent group. This problem brings more challenges to reward estimation, which may be resolved by separating the reward estimation into teammate-agent group reward estimation and opposite-agent group reward estimation, followed by sophisticated reward aggregation. The overall process flow under competitive settings is the same as in cooperative settings (i.e., reward estimation followed by reward aggregation.).
>
>
> > **[4/5] Q4:** It seems like p2p-MARL is the most competitive method to DRE-MARL and also uses reward estimation. Could you clarify the differences between p2p-MARL and DRE-MARL?
>
>
> For Q4, the differences between DRE-MARL and p2p-MARL are as follows: 1) DRE-MARL estimates reward distributions on all action branches while p2p-MARL estimates reward value on one action branch, which is taken by the reward estimator during the training process. 2) DRE-MARL performs reward aggregation after multi-action-branch reward estimation, while p2p-MARL has no reward aggregation.
>
>
> > **[5/5] Q5:** How did you choose k for the distributional estimation? Was there a significant difference between different values of k?
>
>
> In the environment with discrete action space, the K is equal to the number of available discrete actions such as “move forward”, “move backward”, “move left”, “move right”, and “motionless” in MPE. When the environment is fixed, the value of K is determined at the same time.

---

> > ### Author Response · Authors · 2022-08-02
> > **The response to Reviewer oHz3.**
> >
> > ### [III]. Explanation of limitations
> >
> > > **[1/1] L1:** I think that the paper could be improved by further discussion of the limitations of DRE-MARL. As far as I can tell only one limitation is briefly mentioned on Page 9, which is that DRE-MARL and reward aggregation are only used in discrete action spaces. It would be good to get further clarity on the following from the authors:
> > > - How could DRE-MARL be applied to continuous action spaces?
> > > - What limitations does DRE-MARL have that can inspire future work?
> >
> >
> > In our method, the number of action branches matches the number of available discrete actions, so we can set the value of K to be equal to the number of available discrete actions such as “move forward”, “move backward”, “move left”, “move right”, and “motionless” in MPE. But in continuous action space, for example, we want to manipulate a robot arm. The “grasping” action needs us to assign a continuous value such as rotation angle to the robot arm. Right now, the available action is a range, so we can not define the number of K.
> > - One possible solution may be the discretization of the range of the available action value. More sophisticated discretization will bring better manipulation, but it will consume more computational resources at the same time. Although coarse discretization reduces the consumption of physical time, it may hurt the performance.
> > - Another possible solution is learning a network with actions as inputs, just like how we convert discrete-action DQN into continuous-action Q network in DDPG.
> >
> > In the revised version, we add the above discussion into Section 7 of our paper.
> > We hope the above analysis could provide some inspiration and encourage future works to develop novel reward estimation methods.
> >
> > ---
> >
> > We again thank Reviewer oHz3 for reviewing our paper and giving suggestions. We hope our answers have addressed all the concerns the Reviewer has. If so, we would greatly appreciate it if Reviewer oHz3 could consider raising their score. Please let us know if there are more questions.
> >
> > Paper2649 Authors

---

> > > ### Comment · Reviewer_oHz3 · 2022-08-06
> > > **Reply to Author Response**
> > >
> > > The authors have generally addressed my questions and feedback. I would have preferred a more detailed discussion on limitations and future work, but understand there likely isn't enough space for it given the current content of the paper.

---

> > > > ### Author Response · Authors · 2022-08-06
> > > > **Thanks**
> > > >
> > > > We appreciate the reviewer for the valuable suggestions!
> > > > Although we can not discuss more content about the limitations due to space limitations, we will put the discussion about the limitations and future work in the appendix.
> > > >
> > > > Thank you again for your careful review and helpful comments!

---

### Official Review · Reviewer_c9dW · 2022-07-11

**Rating:** 6
**Confidence:** 3
**Soundness:** 3 good
**Presentation:** 3 good
**Contribution:** 3 good

**Summary:**

Considering that high reward uncertainty remains a problem, the authors propose a novel distributed reward estimation framework to enhance multi-gent reinforcement learning. The training process is stabilized by designing multi-action-branch reward estimation and policy-weighted reward aggregation. Multi-action branch reward estimation is first employed to model the reward distribution of all action branches, and then reward aggregation is used to obtain a stable update signal during training. And its effectiveness is demonstrated experimentally.

**Questions:**

* Shouldn't this work be discussed and compared with the related work of Distributional MARL? For example.
    * DFAC Framework: Factorizing the Value Function via Quantile Mixture for Multi-Agent Distributional Q-Learning
    * A Distributional Perspective on Value Function Factorization Methods for Multi-Agent Reinforcement Learning
    * RMIX: Learning Risk-Sensitive Policies for Cooperative Reinforcement Learning Agents


**Ethics Review Area:**

["I don’t know"]

**Strengths And Weaknesses:**

Strengths.
* paper writing is well understood
* The pictures drawn can help understanding
* The motivation is clear and seems to make sense
* table-1 looks like a lot of experiments were done
* Ablation experiments also look interesting


Weaknesses.
* Lack of discussion and experimental comparison with work related to Distributional MARL

---

> ### Author Response · Authors · 2022-08-02
> **The response to Reviewer c9dW.**
>
> We are particularly encouraged that the reviewer finds our method novel and effective. We appreciate the valuable feedback of Reviewer c9dW and respond to the weaknesses and questions below.
>
> ### [I]. Explanation of strengths and weaknesses
>
> > **[1/1] W1:** Lack of discussion and experimental comparison with work related to Distributional MARL
>
>
> We merge the responses to weaknesses and questions and put the discussion of differences and similarities in the following part.
>
>
>
> ### [II]. Explanation of questions
>
> > **[1/1] Q1:** Shouldn't this work be discussed and compared with the related work of Distributional MARL? For example.
> > - DFAC Framework: Factorizing the Value Function via Quantile Mixture for Multi-Agent Distributional Q-Learning
> > - A Distributional Perspective on Value Function Factorization Methods for Multi-Agent Reinforcement Learning
> > - RMIX: Learning Risk-Sensitive Policies for Cooperative Reinforcement Learning Agents
>
>
> The above three articles you recommended are good papers. They all focus on solving the MARL problem from the distributional perspective and all make their efforts to push a step forward.
> The similarity between the above papers and our paper is that we all propose solving the uncertainty problem from the distributional perspective.
> The differences between the above models and our models are shown in the following aspects:
> - They focus on the Q-value uncertainty of the environment and model the distribution on state-action value Q. In contrast, we focus on the reward uncertainty of the environment and model the distribution of reward R.
> - They consider the distributions of all agents by unique mixture methods such as quantile mixture in DFAC. In contrast, we consider the distributions of all action branches by reward aggregation.
>
> In the revised version, we add the above discussion into Section 2 of our paper.
>
> Due to the time limitations, we just run the DFAC at the CN-3 scenario. The results are shown in Table 3. We conduct the experiment in CN-3 with DFAC variants which is based on SC II originally, and adopt the original hyperparameters of DFAC variants. The performance of DFAC variants is not good because we do not adjust the hyperparameters carefully due to time limitations. The DFAC-diql(128) denotes the number of hidden neural is 128.
>
>
>
>
> **Table 3:** Performance comparison of DRE-MARL variants and DFAC variants.
> | model | performance |
> | -------- | -------- |
> | DRE-MARL with  $l_{SS}+g_{SS}$  | -272.7±25.22 |
> | DRE-MARL with  $l_{SMO}+g_{MO}$  | -235.1±16.38 |
> | DRE-MARL with  $l_{MO}+g_{MO}$  | -242.1±16.15 |
> | DRE-MARL with  $l_{MO}+g_{SS}$  | -252.6±17.49 |
> | DFAC-diql(128)  | -786.9±180.2 |
> | DFAC-diql(256)  | -1117.9±21.38 |
> | DFAC-dmix(128)  | -1121.6±21.51 |
>
>
> ---
>
> We again thank Reviewer c9dW for reviewing our paper and giving suggestions. We hope our answers have addressed all the concerns the Reviewer has. If so, we would greatly appreciate it if Reviewer c9dW could consider raising their score. Please let us know if there are more questions.
>
> Paper2649 Authors

---

> > ### Comment · Reviewer_c9dW · 2022-08-03
> > **Reply To The Authors**
> >
> > Dear Authors.
> >
> > The discussion of the differences with the currently available methods of Distributional MARL does not look sufficient, please authors to discuss further in depth the differences with them.
> >
> > As the authors further reported the results, I am curious about the difference in performance between Distributional MARL and DRE-MARL, and whether the authors need to conduct more experiments in the future to support the benefits of DRE-MARL?

---

> > > ### Author Response · Authors · 2022-08-04
> > > **The response to Reviewer c9dW.**
> > >
> > > We thank Reviewer c9dW for the prompt reply and questions. Below we further analyze the differences between our method and the mentioned distributional MARL methods [1,2,3] and the possible reasons that the performance of our method is better than DFAC variants in our preliminary experiments.
> > >
> > > In part I, we first analyze the essential differences between our method and the currently available distributional RL in depth. Next, we explain the reasons behind the experimental phenomenon in Part II. Finally, we put a detailed comparison to [1], [2], and [3] in Part III.
> > >
> > >
> > > ### I. Major differences between our method and distributional MARL methods
> > > **1. Q distribution v.s. reward distribution.**
> > > The mentioned distributional MARL approaches [1,2,3] follow the idea of distributional RL in single-agent problems and learn the distribution of Q values. However, different from a single-agent environment, a MARL environment can be highly stochastic and non-stationary, where uncertainty may exist in the per-timestep rewards of every agent. In this case, although the distribution of Q values can capture the uncertainty of the total future return, it may **(1)** lose some information of per-step reward uncertainty since all future stepwise rewards are blended, and **(2)** require more samples to be accurately estimated due to the high uncertainty. In contrast, our proposed DRE-MARL models the distribution of the per-step reward, which can better capture the per-step uncertainty and is easier to estimate from samples. Therefore, the distributional MARL methods [1,2,3] are more suitable for environments with relatively sparse reward where long-term consideration is more crucial; our DRE-MARL is more suitable for environments with denser reward and higher per-step stochasticity.
> > >
> > >
> > > **2. Quantile mixture v.s. reward aggregation.**
> > > [1] adopts the quantile mixture method to consider all agents' return distributions (See Theorem 3 in [1]). In addition, [2] not only adopts the quantile mixture method but also 1D convolution to consider all agents' return distributions, which realize two variants of DFAC. In contrast, we adopt reward aggregation to consider all reward distributions on all action branches at every step. [1] and [2] use the return distributions to make the decision where inaccurate estimation may directly affect the quality of action sampling. In our method, the aggregated reward is not used to make the decision directly because we have a specific actor network, but is used to update the critic. Then we use the updating signals produced by the critic to perform policy training, where the influence of estimation error is alleviated by policy-weighted reward aggregation. [3] considers all agents' return distributions with dynamic risk-level-masked CVaRs and the mix network, which is different from [1], [2], and our method.
> > >
> > >
> > >
> > >
> > > ### II. Empirical comparison and analysis of performance
> > >
> > > **(1)** The results in Table 3 show that *our DRE-MARL is much better than DFAC-diql(128), DFAC-diql(256), and DFAC-dmix(128) [1] in the MPE environment CN-3*. The potential results are:
> > > - MPE environments have relatively dense rewards, and the per-timestep reward tends to be greatly influenced by the other agents' behaviors (e.g., collision). In this case, as we explained in I, it would be hard to model the long-term Q distribution, and the Q distribution may lose some information of uncertainty about the per-timestep reward.
> > > - QMIX is based on value decomposition, which has been shown effective in SC II in literature [4]. However, in the MPE environments we consider, value decomposition performs not very well. The potential cause is that the reward of SC II is more structured than MPE. For example, the total damage to enemies is the summation of each individual agent.
> > >
> > > **(2)** *We are running more experiments on MPE baselines* in other scenarios with tuned hyperparameters. Our method still outperforms these baselines, and we will update the table once the results are out.
> > >
> > > **(3)** We are not sure what the reviewer was referring to by "more experiments". We would appreciate it if the reviewer could provide some examples. We are happy to conduct new experiments as suggested.

---

> > > > ### Author Response · Authors · 2022-08-04
> > > > **The response to Reviewer c9dW.**
> > > >
> > > > ### III. Detailed comparison with each paper
> > > >
> > > > - **Comparison with [1]**
> > > >
> > > > [1] proposes the Mean-Shape Decomposition method and quantile mixture in value decomposition, bridging the gap between distributional RL and value function factorization methods and enhancing the performance. Compared to this work, our focus is different in the following aspects:
> > > >
> > > > **(1)** From the centralized training perspective: [1] adopts the value factorization method, and the centralized critic is the Q network. DRE-MARL adopts the actor-critic method, and the centralized critic is a V network. Besides, [1] needs to satisfy the assumption of distributional individual-global-max due to the property of value function factorization, while we do not need any assumptions.
> > > >
> > > > **(2)** From the modeled distribution perspective: [1] regards the long return Q value as a random variable Z and models the distribution of Z with the implicit quantile network (IQN), which uses state, action, and quantile sample $\omega$ as input and outputs the corresponding quantile value $Z_{\omega}$. Our paper considers the per-timestep reward as a random variable and proposes multi-action-branch reward estimation to model the distributions.
> > > >
> > > > **(3)** From the mixture method perspective: [1] adopts the quantile mixture method (details can be found in Section 2.7 and Section 3.3 in [1]), which obtains the total return by weighted summation $Z=\sum_{k\in\mathbb{K}}\beta_{k}Z_{k}$ with respect to all agents. We adopt policy-weighted reward aggregation for each agent, which obtains the aggregated rewards with respect to all action branches.
> > > >
> > > > - **Comparison with [2]**
> > > >
> > > > Based on [1], [2] extends the implementation of DFAC variants. Specifically, 1) [2] proposes to fit the individual utilities with the thought of C51. 2) [2] proposes to fit the individual utilities with the quantile function and considers combining the shape of these individual utilities with the quantile mixture.
> > > > Our work is different from this work because:
> > > >
> > > > **(1)** [2] mainly follows [1]. The main difference between [2] and [1] is the implementation: [1] implements the DFAC on VDN and QMIX, obtaining two implementations, DDN and DMIX, while [2] implements two DFAC variants based on QMIX by adopting C51 and IQN to approximate the individual utilities. So it is obvious that the primary differences between DRE-MARL and [2] are the same as those between DRE-MARL and [1].
> > > >
> > > > **(2)** Another difference is that [2] adopts C51 to approximate the individual utilities and 1D convolution to combine all agents’ utilities. While we estimate reward distributions on the multi-action branches and use reward aggregation to combine all predicted reward distributions.
> > > >
> > > > - **Comparison with [3]**
> > > >
> > > > [3] proposes to parameterize the return distributions by a mixture of Dirac Delta functions which are used to calculate Conditional Value at Risk (CVaR). Besides, [3] further introduces dynamic risk level prediction for calculating CVaR and makes risk-sensitive decisions with the calculated CVaRs.
> > > > The differences between [3] and our method are as follows.
> > > >
> > > > **(1)** [3] approximates the long-term return distributions by a mixture of parameterized Dirac Delta functions. We directly model the per-timestep reward distributions on all action branches with the Gaussian reward model.
> > > >
> > > > **(2)** [3] adopts the mix network to consider all agents’ CVaRs while we adopt reward aggregation to consider all action branches’ rewards.
> > > >
> > > > **(3)** [3] mainly considers the risk of actions by computing CVaR based on the value distributions. The goal of [3] is to learn a risk-sensitive policy, but we aim to maximize the cumulative reward by modeling the reward uncertainty.
> > > >
> > > > - **Different ways of parameterization.**
> > > >
> > > > [1] adopts the implicit quantile network to model the return distributions (See Section 2.6 in [1]), while we adopt the reward network to model the per-timestep reward distributions. Besides, [2] adopts C51 to model the probability mass functions of individual utilities andutilizes IQN to approximate the individual untilities, which is also different from us. [3] parameterizes the return distributions with a mixture of Dirac Delta functions, which is different from [1], [2], and our method.
> > > >
> > > >
> > > >
> > > > References:
> > > >
> > > > [1] Sun W F, Lee C K, Lee C Y. DFAC framework: Factorizing the value function via quantile mixture for multi-agent distributional q-learning[C]//ICML. PMLR, 2021: 9945-9954.
> > > >
> > > > [2] Sun W F, Lee C K, Lee C Y. A Distributional Perspective on Value Function Factorization Methods for Multi-Agent Reinforcement Learning[C]//AAMAS. 2021: 1671-1673.
> > > >
> > > > [3] Qiu W, Wang X, Yu R, et al. RMIX: Learning risk-sensitive policies for cooperative reinforcement learning agents[J]. NeurIPS, 2021, 34: 23049-23062.
> > > >
> > > > [4] Rashid T, Samvelyan M, Schroeder C, et al. Qmix: Monotonic value function factorisation for deep multi-agent reinforcement learning[C]//ICML. PMLR, 2018: 4295-4304.

---

### Official Review · Reviewer_JEsw · 2022-07-11

**Rating:** 5
**Confidence:** 2
**Soundness:** 3 good
**Presentation:** 3 good
**Contribution:** 3 good

**Summary:**

Reward uncertainty is a longstanding problem in multi-agent reinforcement learning, which stems from two aspects: the natural uncertainty in the MDP environment, and the actions of other agents. To deal with the reward uncertainty problem, this paper proposes a new framework for estimating the reward distribution and aggregating the estimated reward distribution. Then the proposed method uses the aggregated rewards to update the centralized critic network and the decentralized actor networks. Experiments in the particle domains demonstrate the effectiveness of the proposed method.

**Questions:**

Section 6.3 has conducted ablation studies on different reward estimation methods. I wonder what will happen when no reward estimation is conducted (just using the external rewards to update actor and critic networks). This experiment could better demonstrate the utility of reward estimation in multi-agent reinforcement learning.

This paper refers to the proposed framework as “two-stage” learning (Line 162). Is the reward estimation conducted before the reward estimation? If so, where does the data for estimating the reward distribution come from? Or those two stages are conducted concurrently? This point should be clarified.

Why is the regularization term $L_R$ needed in the reward estimation objective？ What if this term is removed?


**Limitations:**

The authors have discussed the possible limitations of their work in the last paragraph of the submission.

**Strengths And Weaknesses:**

To my best knowledge, the idea of estimating the reward distribution and using the aggregated rewards to optimize the policy is novel, but I get no sense how significant reward estimation is in the multi-agent reinforcement learning domain. Ablation studies on removing the reward estimation module could answer this doubt.

The writing of this paper is mostly clear. In addition, this paper has conducted thorough comparative experiments to compare with state-of-the-art multi-agent algorithms in the particle experiment domain. A more challenging multi-agent benchmark is Starcraft 2. The authors are encouraged to do more experiments in this domain.

---

> ### Author Response · Authors · 2022-08-02
> **The response to Reviewer JEsw.**
>
> We are particularly encouraged that the reviewer finds our method novel and effective. We appreciate the valuable feedback of Reviewer JEsw and respond to the weaknesses, questions, and limitations below.
>
> ### [I]. Explanation of strengths and weaknesses
>
> > **[1/1] W1:** To my best knowledge, the idea of estimating the reward distribution and using the aggregated rewards to optimize the policy is novel, but I get no sense how significant reward estimation is in the multi-agent reinforcement learning domain. Ablation studies on removing the reward estimation module could answer this doubt.
> The writing of this paper is mostly clear. In addition, this paper has conducted thorough comparative experiments to compare with state-of-the-art multi-agent algorithms in the particle experiment domain. A more challenging multi-agent benchmark is Starcraft 2. The authors are encouraged to do more experiments in this domain.
>
>
>
> StarCraft II (SCII) is indeed a more challenging multi-agent benchmark. We are happy to evaluate our method in SCII, but it is hard to get results within the short rebuttal period. We will conduct the investigation of more challenging tasks such as SCII and put the experiments of SCII in future works.
>
>
>
> ### [II]. Explanation of questions
>
> > **[1/3] Q1:** Section 6.3 has conducted ablation studies on different reward estimation methods. I wonder what will happen when no reward estimation is conducted (just using the external rewards to update actor and critic networks). This experiment could better demonstrate the utility of reward estimation in multi-agent reinforcement learning.
>
>
> For Q1, due to time limitations, we evaluate the performance of our model with just external reward signals on CN-3. **The experimental results are shown in the last line of Table 2** (in the answer to Q3), which shows that **only using external rewards indeed will hurt the performance**.
>
>
>
> > **[2/3] Q2:** This paper refers to the proposed framework as “two-stage” learning (Line 162). Is the reward estimation conducted before the reward estimation? If so, where does the data for estimating the reward distribution come from? Or those two stages are conducted concurrently? This point should be clarified.
>
>
> For Q2, we do the reward estimation and the reward aggregation **sequentially in every training epoch** rather than doing all the reward estimation at once before all the reward aggregation. The data set used to estimate reward distributions comes from the replay buffer. As illustrated in Algorithm 1, a more intuitionistic but detailed process of one training epoch is as follows: 1) We interact with the environment and deposit the transitions in the replay buffer. 2) We sample a batch of transitions B from the replay buffer. 3) These transitions will be used to train reward estimators first. Then we again use trained reward estimators to predict reward distributions $\hat{R}$ of all action branches with states that are stored in transitions B. 4) We sample rewards $\hat{r}$ from $\hat{R}$ and deposit $\hat{r}$ and environmental rewards that are stored in transitions B together. 5) Perform reward aggregation with Equation 2 and Equation 5. 6) Update the critic and the actors.
>
>
>
> > **[3/3] Q3:** Why is the regularization term $L_R$ needed in the reward estimation objective？ What if this term is removed?
>
>
> For Q3, we add the regularization term $L_R$ out of consideration for training stability. If this term is removed, the policy learning process may be influenced. For example, a larger variance of $\pmb{\mu}$ will make the aggregated rewards more fiercely, affecting the critic's updating. A more significant $\sigma$ value will lead to a smaller gradient of Equation 1, which will affect the updating of the reward estimators. To illustrate the effect of $L_R$, we evaluate our model without $L_R$ on the CN3 scenario, and the results are shown in Table 2. From Table 2, we can see that **removing the regularization term $L_R$ will hurt the performance compared with the corresponding "with $L_R$" models**.

---

> > ### Author Response · Authors · 2022-08-02
> > **The response to Reviewer JEsw.**
> >
> > **Table 2:** Performance comparison of DRE-MARL with and without regularization term $L_R$ while training with the team rewards and evaluating without the r_ac−dist setting.
> > The values represent mean episodic rewards.
> > | comparison items | model |
> > | -------- | -------- |
> > | $l_{SS}+g_{SS}$ with $L_R$   | **-272.7±25.22**  |
> > | $l_{SS}+g_{SS}$ without $L_R$   | -274.6±19.13  |
> > | $l_{SMO}+g_{MO}$ with $L_R$   |  **-235.1±16.38** |
> > | $l_{SMO}+g_{MO}$ without $L_R$   | -365.0±25.19  |
> > | $l_{MO}+g_{MO}$ with $L_R$   | **-242.1±16.15**  |
> > | $l_{MO}+g_{MO}$ without $L_R$   |  -258.9±17.36 |
> > | $l_{MO}+g_{SS}$ with $L_R$  | **-252.6±17.49**  |
> > | $l_{MO}+g_{SS}$ without $L_R$   | -354.2±23.47  |
> > | no reward estimation  | -258.2±20.75  |
> >
> >
> >
> >
> > ### [III]. Explanation of limitations
> >
> > > **[1/1] L1:** The authors have discussed the possible limitations of their work in the last paragraph of the submission.
> >
> >
> > We have discussed the limitations of our method in Section 7, and these problems will be investigated in future works.
> >
> > ---
> >
> > We again thank Reviewer JEsw for reviewing our paper and giving suggestions. We hope our answers have addressed all the concerns the Reviewer has, especially the ablation of the regularization term $L_R$ and no reward estimation. If so, we would greatly appreciate it if Reviewer JEsw could consider raising their score. Please let us know if there are more questions.
> >
> > Paper2649 Authors

---

> ### Author Response · Authors · 2022-08-08
> **Follow-up to Reviewer JEsw**
>
> Dear Reviewer JEsw,
>
> We appreciate the reviewer's positive feedback and worthy suggestions for our paper. Furthermore, the recommendations of ablation studies and the clarification of our framework help us improve the quality of our paper further. As the end of the discussion is approaching, we are wondering if there are any additional potential clarifications or suggestions that you think would help us improve this manuscript.
>
> Following your recommendations, we provided ablation studies (as shown in Table 2) about the reward estimation and the regularization term, which were also added to the paper. Besides, we also explained the relationship between reward estimation and reward aggregation in detail to make it clear to you and other readers.
>
> We thank the reviewer's effort in the review of our paper. We hope all the concerns have been addressed. Please let us know if there are more questions. We are happy to address any further questions or concerns.
>
> Thank you again for your careful review and helpful comments!
>
> Kind regards,
>
> Paper2649 Authors

---

### Official Review · Reviewer_TJjg · 2022-07-11

**Rating:** 6
**Confidence:** 3
**Soundness:** 3 good
**Presentation:** 3 good
**Contribution:** 3 good

**Summary:**

This paper focus on addressing the problem of high reward uncertainty in multi-agent reinforcement learning (MARL). To address this problem, this paper proposes a new Distributional Reward Estimation (DRE) framework, which is composed of multi-action-branch reward estimation and policy-weighted reward aggregation for stable training. The full method, DRE-MARL is built on top with the architecture of centralized training and decentralized evaluation (CTDE), which consists of N decentralized actors and a centralized critic. Empirically, the proposed method is evaluated in several MARL environments, i.e. cooperative navigation, reference, and treasure collection with different number of agents. Compared with other SOTA MARL methods, DRE-MARL achives the best performance in most of the environments using three types of reward settings.

**Questions:**

1. Line 55, "If we can obtain the potential rewards on other action branches, we will perform more stable critic updating and thus achieve better performance." Is there any assumption of requirement on the design of action branches to make this sentence correct? What if the other actions are opposite the the current action of a specifict agent?

2. What is the motivation for the design of different aggregation weights?

3. DRE-MARL requires individual reward estimator for each agent. How muc is the computational cost gain of DRE-MARL compared with other SOTA-MARL methods?

**Limitations:**

The authors have discussed the limitations and broader impacts of their work in section 7.

==Limitations==
1. DRE-MARL requires a prior assumption about the form of reward distribution. Although the distribution usually can be chosen arbitrarily, it may hurt performance when we choose a very complex distribution.
2. Besides, reward aggregation is only used in discrete action space. If would be better if the method can be extented to more general continuous action space.

==Broader impacts==
The authors does not see any negative societal impacts of this work while using the proposed method in practice.


**Strengths And Weaknesses:**

==Originality==

The proposed DRE-MARL is novel and interesting. The main idea is to develop distributional reward estimation followed by policy-weighted reward aggregation for MARL. This idea is intuitively similar to human's decision making process that considers all possible consequences of all action branches. The connections and differences of this work and previous work are well discussed and the related work are cited in the paper.


==Quality==

The proposed method is technically sound. The experiments are conducted in diverse settings with ablations, and the results well supports the claims. Although there are some limitations, this work is a complete piece of work in addressing the reward uncertainty problem in MARL.

==Clarity==

This paper is clearly written and well organized. The problem is well formulated with a clear intruduction of the cause of reward uncertainty. The method is clearly motivated and introduced. The limitations and potential impacts are also discussed.

==Significance==

This paper provides a novel distributional reward estimation method to address the high reward uncertainty in multi-agent reinforcement learning. The proposed method is novel and can be applied in other MARL methods.

---

> ### Author Response · Authors · 2022-08-02
> **The response to Reviewer TJjg.**
>
> We are particularly encouraged that the reviewer finds our method novel and effective. We appreciate the valuable feedback of Reviewer TJjg and respond to the questions and limitations below.
>
> ### [I]. Explanation of questions
>
> > **[1/3] Q1:** Line 55, "If we can obtain the potential rewards on other action branches, we will perform more stable critic updating and thus achieve better performance." Is there any assumption of requirement on the design of action branches to make this sentence correct? What if the other actions are opposite the the current action of a specifict agent?
>
>
> We express our gratitude for your comments, and we will revise this sentence to avoid potential ambiguity. We want to clarify that we actually do not need extra assumptions. Although inaccurate reward estimation may exist in certain action branches during training, our model possesses relative robustness due to the policy-weighted reward aggregation. Because the strategy of policy-weighted reward aggregation puts larger weights on some action branches where the corresponding actions will also be performed more frequently during the interaction, so we can obtain more samples and be more confident that the reward estimation is relatively accurate in practice. As for the second doubt of Q1, “What if the other actions are opposite the current action of a specific agent?”, we do not see any specific scene that matches the foregoing situation. Despite the fact that the opposite actions may appear, we can still obtain the corresponding rewards on that opposite action branch as long as the policy endows them with a certain probability. We have revised this sentence in the revised edition.
>
>
> > **[2/3] Q2:** What is the motivation for the design of different aggregation weights?
>
>
> The reason for adopting policy-weighted reward aggregation is that the policy-weighted reward aggregation strategy can better reflect the current policy's expected return, enabling the agent to evaluate its historical experiences thoughtfully. Additionally, in Equation 2 and Equation 5, we adopt various aggregation methods to evaluate our model in different and frequently-used aggregation settings, which enables us to assess our method comprehensively in the experiments.
>
>
> > **[3/3] Q3:** DRE-MARL requires individual reward estimator for each agent. How much is the computational cost gain of DRE-MARL compared with other SOTA-MARL methods?
>
>
> In order to evaluate the computational cost of our model, we compare it with other models from three aspects: 1) parameters, 2) physical training time (min), and 3) max memory consumption (MB), where we subdivide parameters into total parameters, trainable parameters, and untrainable parameters. The testing is conducted on the CN-3 scenario.
> The physical training time (min) is tested based on CPUs without GPUs, and we run the codes with one thread. The type of CPU is Intel(R) Xeon(R) Gold 6230 CPU @ 2.10 GHz.
> The results are shown in Table 1, which shows that the computational cost of our model is competitive or even more efficient in practice, such as the number of trainable parameters.
>
>
> **Table 1:** The comparison of the computational cost of different models based on the CN-3 scenario. The items of the comparison contain three aspects: parameters, physical training time (min), and max memory consumption (MB).
> | comparison items | DRE-MARL (ours) | MAPPO | MAAC | QMIX | MADDPG | IQL
> | -------- | -------- | -------- |-------- | -------- |-------- | -------- |
> | total  parameters   | 198440 | 72854 | 450364| 307330|172884 |564642 |
> | trainable  parameters | 54126   | 72854  | 450364 | 123488 | 86436 | 141147 |
> | untrainable  parameters | 144314   | 0  | 0 | 183842 | 86448 | 423495 |
> | physical training time (min) | 256.0±1.859   | 336.0±9.623  | 915.9±6.270 | 867.1±16.54 | 166.9±1.976 | 217.4±2.311 |
> | max memory consumption (MB) | 1354.±60.18   | 310.6±9.623  | 229590.1±65.69 | 2146.5±22.95 | 3754.±227.3 | 668.4±20.18 |

---

> > ### Author Response · Authors · 2022-08-02
> > **The response to Reviewer TJjg.**
> >
> > ### [II]. Explanation of limitations
> >
> > > **[1/1] L1:** The authors have discussed the limitations and broader impacts of their work in section 7.
> > DRE-MARL requires a prior assumption about the form of reward distribution. Although the distribution usually can be chosen arbitrarily, it may hurt performance when we choose a very complex distribution.
> > Besides, reward aggregation is only used in discrete action space. If would be better if the method can be extented to more general continuous action space.
> >
> >
> > For limitations, we have discussed them in Section 7 of this paper. We admit these limitations indeed hinder our model from being used in some other scenes, such as continuous action space. However, these problems may be resolved in the future. For example, one can discretize the range of action value in a continuous action space. The assumption of reward distribution limitation may be solved by using a cluster of basic distributions as reward distribution.
> >
> > ---
> >
> > We again thank Reviewer TJjg for reviewing our paper and giving suggestions. We hope our answers have addressed all the concerns the Reviewer has. If so, we would greatly appreciate it if Reviewer TJjg could consider raising their score. Please let us know if there are more questions.
> >
> > Paper2649 Authors

---

> ### Author Response · Authors · 2022-08-08
> **Follow-up to Reviewer TJjg**
>
> Dear Reviewer TJjg,
>
> We appreciate the reviewer's constructive suggestions for our paper. Your comments help us increase the readability and quality of the manuscript, and we also improve our paper according to other reviewers' comments. As the end of the discussion is approaching, we are wondering if there are any additional potential clarifications or suggestions that you think would help us improve this manuscript.
>
> Following your suggestions, we provided the clarifications and explanations for the assumption of action branches, the motivation of aggregation weights, and the comparison of computational costs. Additionally, we compared our model and other baselines from the perspectives of parameters, physical training time, and memory consumption. The comparison is shown in Table 1.
>
> We thank the reviewer's effort in the review of our paper. We hope all the concerns have been addressed. Please let us know if there are more questions. We are happy to address any further questions or concerns.
>
> Thank you again for your careful review and helpful comments!
>
> Kind regards,
>
> Paper2649 Authors

---

### Meta-Review · Area_Chair_FCPk · 2022-08-25

**Recommendation:** Accept
**Confidence:** Certain

**Metareview:**

The reviewers carefully analyzed this work and agreed that the topics investigated in this paper are important and relevant to the field. They believe that the NeurIPS community could benefit from the ideas and techniques presented in this work. They argued, e.g., that the paper is novel and interesting, technically sound, clearly written, and that the method is clearly motivated and introduced. One reviewer expressed a few technical concerns, to which the authors responded appropriately. The authors have also, post-submission, further compared their model and other baselines from different perspectives. One reviewer pointed out that a limitation of the paper is the lack of discussion and experimental comparison with other work related to Distributional MARL. The authors responded to this, but the reviewer requested further details and a more thorough discussion; the authors then expanded their initial response via two detailed rebuttal messages, which were considered to be satisfactory. Finally, another reviewer (who also expressed positive views on this work) mentioned that the authors could have provided more details on the limitations of their method. Overall, all reviewers were positively impressed with the quality of this work and look forward to an updated version of the paper that addresses the suggestions mentioned in their reviews.

**Award:**

No

---

### Decision · Program_Chairs · 2022-09-14

Accept